# Natural selection and repeated patterns of molecular evolution following allopatric divergence

Yibo Dong[1,2†], Shichao Chen[3,4,5†], Shifeng Cheng[6], Wenbin Zhou[1], Qing Ma[1], Zhiduan Chen[7], Cheng-Xin Fu[8], Xin Liu[6*], Yun-peng Zhao[8*], Pamela S Soltis[3*], Gane Ka-Shu Wong[6,9,10*], Douglas E Soltis[3,4], Qiu-Yun(Jenny) Xiang[1*]

[1]Department of Plant and Microbial Biology, North Carolina State University, Raleigh, United States; [2]Plant Biology Division, Noble Research Institute, Ardmore, United States; [3]Florida Museum of Natural History, University of Florida, Gainesville, United States; [4]Department of Biology, University of Florida, Gainesville, United States; [5]School of Life Sciences and Technology, Tongji University, Shanghai, China; [6]Beijing Genomics Institute, Shenzhen, China; [7]State Key Laboratory of Systematic and Evolutionary Botany, Institute of Botany, Chinese Academy of Sciences, Beijing, China; [8]Laboratory of Systematic & Evolutionary Botany and Biodiversity, College of Life Sciences, Zhejiang University, Hangzhou, China; [9]Department of Biological Sciences, University of Alberta, Edmonton, Canada; [10]Department of Medicine, University of Alberta, Edmonton, Canada

**\*For correspondence:**
liuxin@genomics.cn (XL);
ypzhao913@gmail.com (Y-Z);
psoltis@flmnh.ufl.edu (PSS);
gane@ualberta.ca (GK-SW);
dsoltis@ufl.edu (DES);
jenny_xiang@ncsu.edu (Q-Y(J)X)

[†]These authors contributed equally to this work

**Competing interests:** The authors declare that no competing interests exist.

**Abstract** Although geographic isolation is a leading driver of speciation, the tempo and pattern of divergence at the genomic level remain unclear. We examine genome-wide divergence of putatively single-copy orthologous genes (POGs) in 20 allopatric species/variety pairs from diverse angiosperm clades, with 16 pairs reflecting the classic eastern Asia-eastern North America floristic disjunction. In each pair, >90% of POGs are under purifying selection, and <10% are under positive selection. A set of POGs are under strong positive selection, 14 of which are shared by 10–15 pairs, and one shared by all pairs; 15 POGs are annotated to biological processes responding to various stimuli. The relative abundance of POGs under different selective forces exhibits a repeated pattern among pairs despite an ~10 million-year difference in divergence time. Species divergence times are positively correlated with abundance of POGs under moderate purifying selection, but negatively correlated with abundance of POGs under strong purifying selection.

## Introduction

Identifying the factors driving divergence is a key research topic of speciation genomics (*Wolf and Ellegren, 2017*). Although there has been progress in identification of individual speciation genes and genomic islands of divergence in speciation (*Nosil and Feder, 2013*; *Renaut et al., 2013*), it remains an open question as to whether there are general genomic patterns and processes of molecular evolution accompanying the divergence of species. Natural selection and drift are recognized as the major processes promoting divergence. However, the relative roles of these processes, the proportion of the genome affected by natural selection, and the strength of selection in generating genomic divergence have remained poorly understood (*Kimura, 1979*; *Noor and Bennett, 2009*; *Barrett and Hoekstra, 2011*; *Nosil and Feder, 2013*). A few recent studies have uncovered repeated patterns of genomic divergence driven by selection in species formation (*Gagnaire et al., 2013*; *Arnegard et al., 2014*; *Renaut et al., 2014*; *Soria-Carrasco et al., 2014*) and suggested that

divergence may often result from changes at a relatively small subset of genes (e.g., *Conte et al., 2012*).

Geographic speciation is a common driving force in generating biodiversity (*Coyne and Allen Orr, 1998*). Given sufficient time and/or selection pressures, genomic divergence between allopatric populations is expected (*Coyne and Allen Orr, 1998*; *Nosil, 2008*). However, how genes diverge over time at a genomic scale after major geographic isolation that completely interrupts gene flow has remained an unanswered question. Meta-analyses of closely related species that span a range of time scales of geographic isolation will be, therefore, particularly beneficial for gaining insights into this question (*Nosil et al., 2009*). Such studies might allow us to reconstruct how genomic divergence unfolds as speciation proceeds through time.

Putatively orthologous (single-copy) genes (POGs) identified in RNA-seq data sets of closely related species offer a valuable genome-wide window on molecular divergence of a unique subset of the genome following speciation (*Feder et al., 2012*; *De Smet et al., 2013*). Investigating how these genes diverge during speciation and subsequent evolution contributes an important perspective to speciation genomics. Such studies may illuminate the functional divergence of the allopatric genomes and the underlying ecological forces that shape them. The forests of eastern Asia (EA) and eastern North America (ENA) share 65 genera of seed plants with closely related species occurring in the two areas (*Li, 1952*; *Boufford and Spongberg, 1983*; *Wu, 1983*; *Wen, 1999*; *Wen et al., 2010*; *Wen et al., 2016*). This well-known floristic disjunction is a major phytogeographic pattern of the Northern Hemisphere and has been known since the time of Linnaeus (*Wen, 1999*). The origins of the disjunction and impact on plant evolution have been investigated over the past 30 years (*Xiang et al., 1998*; *Wen, 1999*; *Xiang et al., 2000*; *Wen, 2001*; *Xiang and Soltis, 2001*; *Donoghue and Smith, 2004*; *Harris et al., 2013*; *Manos and Meireles, 2015*; *Qian et al., 2017*). Many of these genera are small with one to a few species. Previous analyses using mainly plastid and internal transcribed spacer (ITS) DNA sequence markers revealed that the divergence times of disjunct sister species in EA and ENA varied greatly from the Miocene to the Pleistocene or even more recently (15 mya to <2.0 mya), although some disjunct clades in the two areas diverged earlier (*Xiang et al., 2000*; *Wen, 2001*; *Milne and Abbott, 2002*; *Donoghue and Smith, 2004*; *Harris et al., 2013*). These genera therefore represent different stages of allopatric divergence and are ideal for studying genomic divergence associated with geographic speciation.

In this paper, we compare the divergence patterns of genes identified as POGs in leaf transcriptomes of 20 allopatric species/variety pairs (or taxon pairs), all but four of which represent the EA-ENA floristic disjunction (*Table 1*). The 16 EA-ENA disjunct species or subspecies/variety pairs represent diverse clades of flowering genera with two to several species: *Campsis* (two spp.) – Lamiales, *Convallaria* (three vars. of 1 sp.) – Asparagales, *Cornus* (two subclades, *Cornus*-1 with eight spp., *Cornus*-2 with two spp.) – Cornales, *Cotinus* (two spp.) – Sapindales, *Croomia* (three spp.) – Pandanales, *Gelsemium* (three spp.) – Gentianales, *Hamamelis* (six spp.) – Saxifagales, *Liquidambar* (4–15 spp.) – Saxifragales, *Liriodendron* (two spp.) – Magnoliales, *Meehania* (seven spp.) – Lamiales, *Menispermum* (two spp.) – Ranunculales, *Nelumbo* (two spp.) – Proteales, *Penthorum* (two spp.)– Saxifragales, *Phryma* (two spp. or two vars. of 1 sp.) – Lamiales, *Sassafras* (three spp.) – Magnoliales, and *Saururus* (two spp.) – Piperales (*Appendix 1—figure 1*). These EA-ENA disjunct genera are among the classic examples of the EA-ENA floristic disjunction (*Li, 1952*; *Wu, 1983*; *Wen, 1999*) and have one to a few species isolated in the two areas, and in a few cases, a species also occurs in western North America (e.g., *Cornus*-1) and/or southwestern Asia and southeastern Europe (e.g., *Convallaria*, *Liquidambar*, *Nelumbo*). The species pairs we compared in these genera (*Table 1*) span a range of divergence times (*Xiang et al., 2000*; *Wen, 2001*; *Milne and Abbott, 2002*; *Wen et al., 2010*).

As noted, four of the 20 pairs represent other allopatric pairs (i.e., *Acorus*, two spp. – Acorales, *Calycanthus*, three spp. – Laurales, *Dysosma*, seven spp. – Ranunculales). These additional pairs were included in the study to increase our sampling of geographically separated taxon pairs for comparison and to cover a wide window of times, from recent divergence of speciation to further divergence of the descendant species. The species pairs analyzed follow: one EA-Western North American (WNA) pair (*Calycanthus chinensis – C. occidentalis*), one Asia/S. Europe-ENA pair (*Cotinus coggygria – C. obovatus*), one EA-EA/NA pair (*Acorus gramineus – A. calamus*), and one Asian pair (*Dysosma versipellis – D. pleiantha*; Ranunculales) (*Table 1*). Species of *Acorus* and *Dysosma* partially overlap in their geographic ranges, but have diverged in habitat and do not occur together. *Acorus calamus* naturally occurs throughout China and adjacent countries, and in North America in swamps,

**Table 1.** List of biogeographic pairs and number of examined putative single copy orthologs (POGs) (sum from two species), maximum length (MAL), minimum length (MIL), and average length (AL) of POGs, number of POGs mapped to single copy genes in seed plants (NSP) and number of POGs that were mapped to single copy genes in 20 angiosperm plants (NAP).

| Genera | Species | Distribution | #of POGs | MAL of POGs (bp) | MIL of POGs (bp) | AL of POGs (bp) | NSP of POGs | NAP of POGs |
|---|---|---|---|---|---|---|---|---|
| Acorus | A.calamus, A.gramineus | EA-Asia/America | 17192/2 | 6213 | 102 | 874 | 77 | 81 |
| Calycanthus | C.chinensis, C.occidentalis | EA-WNA | 23116/2 | 6705 | 102 | 908 | 96 | 152 |
| Campsis | C.grandiflora, C.radicans | EA-ENA | 20840/2 | 5697 | 102 | 734 | 66 | 112 |
| Convallaria | C.majalis var. keiskei, C.majalisvar. Montana | EA-ENA | 30322/2 | 7041 | 102 | 502 | 53 | 75 |
| Cornus-1 | C.ellipica, C.florida | EA-ENA | 24398/2 | 5682 | 102 | 800 | 154 | 259 |
| Cornus-2 | C.alternifolia, C. controversa | EA-ENA | 26490/2 | 7455 | 102 | 836 | 210 | 323 |
| Cotinus | C.coggygria, C.obovatus | Eurasia-ENA | 26494/2 | 7374 | 102 | 889 | 237 | 361 |
| Croomia | C.japonica,C.paueiflora | EA-ENA | 23160/2 | 6480 | 102 | 817 | 71 | 112 |
| Dysosma | D.pleiantha, D.versipellis | Asia* | 24270/2 | 11694 | 102 | 944 | 132 | 218 |
| Gelsemium | G.elegans, G.sempervirens | EA-ENA | 22774/2 | 6801 | 102 | 980 | 129 | 194 |
| Hamamelis | H.japonica, H.vernalis | EA-ENA | 23052/2 | 7308 | 102 | 955 | 221 | 365 |
| Liquidarnbar | L.styraciflua, L.formosana | EA-ENA | 24844/2 | 7041 | 102 | 893 | 196 | 343 |
| Liriodendron | L.chinense, L.tulipifera | EA-ENA | 23060/2 | 7053 | 102 | 748 | 77 | 120 |
| Meehania | M.fargesii, M.cordata | EA-ENA | 25622/2 | 5562 | 102 | 919 | 97 | 166 |
| Menispermum | M.canadense, M.dauricum | EA-ENA | 27490/2 | 7347 | 102 | 449 | 60 | 121 |
| Nelumbo | N.lutea, N.nucifera | EA-ENA | 23988/2 | 7329 | 102 | 774 | 138 | 220 |
| Penthorum | P.chinense, P.sedoides | EA-ENA | 23288/2 | 15225 | 102 | 882 | 161 | 245 |
| Phryma | P.aleptostachya, P.leptostachya | EA-ENA | 25096/2 | 11445 | 102 | 863 | 106 | 180 |
| Sassafras | S.albidum, S.tzumu | EA-ENA | 25070/2 | 7524 | 102 | 809 | 104 | 153 |
| Saururus | S.cernuus, S.chinensis | EA-ENA | 19386/2 | 6300 | 102 | 888 | 53 | 84 |

*Evergreen-deciduous forests pair.

pond sides, and standing water below 2800 m. *Acorus gramineus* occurs in eastern, southern, and western China, and adjacent countries of EA in dense forests, moist rocky stream banks, and meadows below 2600 m. *Dysosma versipellis* and *D. pleiantha* both occur in eastern, central, and southern China, but *D. versipellis* is found in deciduous forests while *D. pleiantha* occurs in evergreen forests.

Additional information on the 20 taxon pairs and their distributions can be obtained from the *Flora of China* (http://www.efloras.org) and the *Flora of North America* (http://floranorthamerica. org/). The 20 taxon pairs we compared span a range of divergence times (*Xiang et al., 1998*; *Xiang et al., 2000*; *Milne and Abbott, 2002*; *Donoghue and Smith, 2004*; *Wen et al., 2010*;

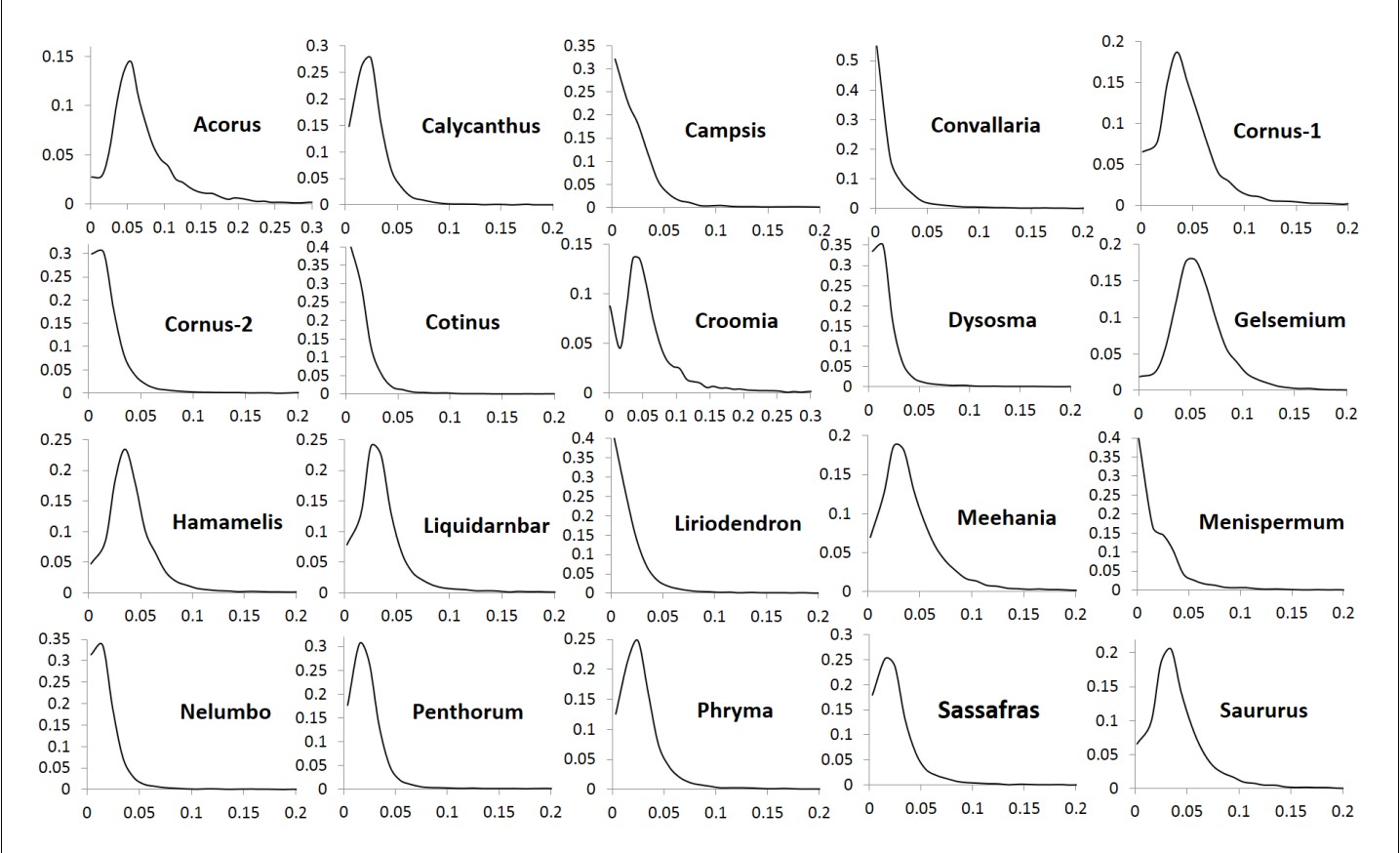

**Figure 1.** Frequency distribution of synonymous substitutions per synonymous site (*Ks*) of putative orthologs (POGs) from leaf transcriptomes of 20 species/variety pairs of angiosperms.

*Harris et al., 2013*; *Wen et al., 2016*), mostly representing sister species or varieties; however, in a few cases, the selected species represent sister clades of one vs. two species (*Calycanthus, Gelsemium, Sassafras*), one vs. a few species (*Hamamelis*), or two or three vs. a few species (*Cornus*-1, *Liquidambar*).

We examined the pattern of variation of synonymous substitutions (*Ks*; the number of synonymous substitutions per synonymous site), nonsynonymous substitutions (*Ka*; the number of nonsynonymous substitutions per nonsynonymous site), and the ratio (*Ka/Ks*) for each POG within each taxon pair using a custom pipeline (*Appendix 1—figure 2*) and compared the patterns among taxon pairs representing these diverse angiosperm lineages. The ratio of nonsynonymous to synonymous nucleotide substitutions (*Ka/Ks*) is probably the most common measure of selection pressure and for identifying positive selection (*Ka/Ks* > 1) (*Yang, 2003*) in molecular evolutionary studies. The method is known to be a conservative approach to call positive selection on a gene as it uses an average *Ka/Ks* ratio across all nucleotide sites. We also examined the relative proportions of genes under different levels of selection forces and identify and annotate genes under strong positive selection based on the *Ka/Ks* values (>2). Finally, we tested if the level of molecular divergence at synonymous sites (as measured by *Ks* at the peak frequency) reflects the relative divergence time of the species pair. Results from such analyses will illuminate the genomic pattern of genetic divergence of these genes under geographic isolation, and may also shed light on a small window of genomic divergence that unfolds as speciation proceeds through time. Furthermore, this study will provide a 'genome-wide' view of evolutionary divergence of species exhibiting the EA-ENA phytogeographic pattern, in comparison to previous understanding based on a few gene regions. The POGs examined in this study represented 52–68% of the total orthologous gene families in the transcriptome data (*Supplementary file 1*).

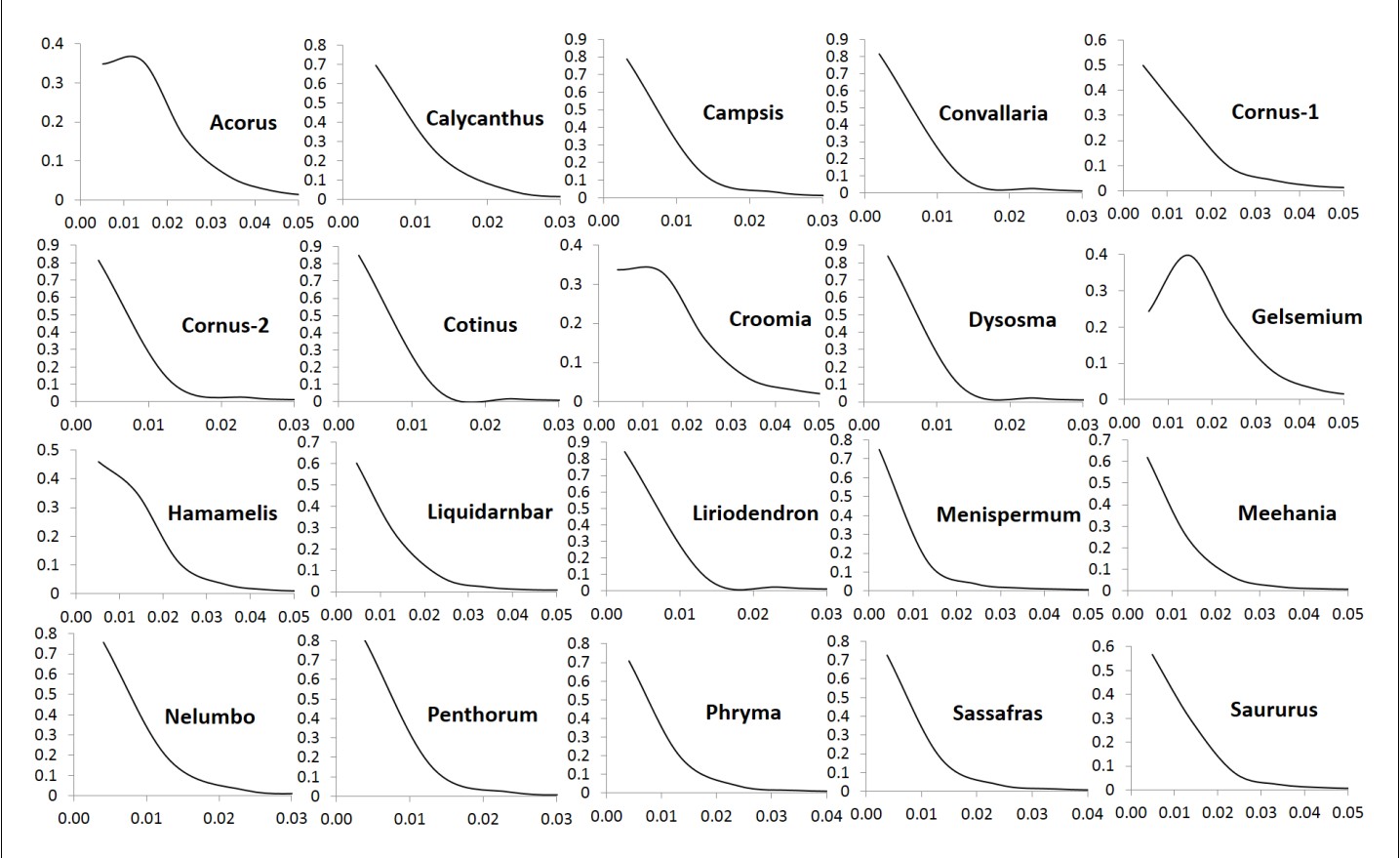

**Figure 2.** Frequency distribution of nonsynonymous substitutions per non-sysnonymous site (*Ka*) of putative orthologs (POGs) of leaf transcriptomes from 20 species/variety pairs of angiosperms.

We stress that the POGs in leaf transcriptomes analyzed here are putative orthologs between the two species of each biogeographic pair found in the transcriptomes by OrthoMCL followed by customized scripts (see Methods below). It is possible, therefore, that some of these genes may have more than one copy in the genome (as a small gene family), but only one of them was expressed to a detectable level in the leaf. In such cases, it is possible that the two single-copy transcripts from the taxon pair may represent paralogous copies of the low-copy gene family, making the gene comparison non-orthologous. However, we think the likelihood of non-orthology of these genes is probably very low because in each case the two taxa are either sister species or intraspecific taxa, or represent the products of recently diverged sister clades where conservation of expression of a gene copy may be expected. We therefore do not think that the low probability of non-homology will affect our conclusions derived from thousands of putatively orthologous comparisons. However, we do want to make it clear that the discussion and conclusions about 'genome-wide' pattern of evolutionary divergence and processes in this paper are restricted to these genes of the genome. Nonetheless these POGs represent a significant portion of the gene families expressed in the transcriptome of each species (52%–68%; *Supplementary file 1*).

Comparison of the rates of nonsynonymous and synonymous substitutions provides a useful tool for understanding the mechanisms of DNA sequence evolution. A ratio of nonsynonymous (*Ka*) to synonymous (*Ks*) nucleotide substitutions greater than one is a common method for identifying positive selection in molecular evolutionary studies (*Yang, 2003*). We are aware that uncertainty in *Ka/Ks* ratios is high when *Ks* is zero or close to zero. This uncertainty may lead to inflation of *Ka/Ks* ratios or false identification of genes under strong positive selection due to a high, infinite, or undefined *Ka/Ks* value. To minimize this problem, we removed genes with *Ks* = 0 and genes with *Ka/Ks* = 99

for all analyses and only focused on genes with $Ka/Ks > 2$ as candidates of strong positive selection for gene annotation analysis. Our detailed examination of the $Ka$ and $Ks$ data (see Discussion below) provided evidence supporting the candidate genes under positive selection detected in the study are unlikely artifacts of small $Ks$. Further, the $Ka/Ks > 2$ ratios in these genes are unlikely a result of saturation of synonymous substitutions. Saturation of synonymous substitutions is often expected at deeper phylogenetic levels. We further used alternative method (*Wagner, 2007*) to check for signals of positive selection in these genes with $Ka/Ks > 2$. This method detects variation clusters of aggregated nucleotide substitutions that are too closely spaced to be observed by chance alone (violating the prediction from neutral evolution). Both the $Ka/Ks$ and variation cluster methods may underestimate the number of genes under positive selection because some genes with $Ka/Ks < 1$ may be under positive selection at only a few sites and these sites may not be necessarily clustered in the gene. However, our goal is not to determine exactly how many POGs are under positive selection, and this caveat does not affect our comparisons of average selection pressures among genes or comparisons of general patterns of gene divergence across species pairs. The genes identified as under positive selection by both methods are likely the true positive selection genes and will provide top candidates for future studies for their functions in driving allopatric divergence in these taxa.

## Results

The number of POGs in the leaf transcriptomes of the 20 biogeographic taxon pairs varies from 8596 to 15161 (*Table 1*). Among these POGs, 2241 are shared by over 50% of the taxa, 333 are shared by over 80% of the taxa, 79 are shared by over 90% of the taxa, and seven are shared by all of the taxa. Most of these genes exhibit low divergence, with synonymous substitution ($Ks$) values less than 0.1 and nonsynonymous substitution values ($Ka$) less than 0.04. Less than 1% of the genes exhibit greater divergence with $Ks$ values greater than 0.15 (*Figure 1*) and $Ka$ values greater than 0.05 (*Figure 2*). The $Ks$ values at peak frequency when plotted in intervals of 0.01 range among the taxon pairs from <0.01 to<0.06 (*Figure 1*). In *Convallaria*, *Liriodendron*, *Cotinus*, *Menispermum*, and *Campsis*, the $Ks$ values at peak frequency are less than 0.01, while in *Cornus*-2, *Dysosma*, *Nelumbo*, *Penthorum*, and *Sassafras*, they are between 0.01 and 0.02. In *Calycanthus*, *Liquidambar*, *Meehania*, *Menispermum*, and *Phryma*, they are between 0.02 and 0.03, and in *Cornus*-1, *Croomia*, *Hamamelis*, and *Saururus*, they are between 0.03 and 0.04. The largest values are observed in *Acorus* and *Gelsemium*, between 0.05 and 0.06 (*Supplementary file 2*). A portion of the POGs, 53–237 among the genera, were aligned to putative single-copy or low-copy genes shared by seed plants (*Li et al., 2017*), and 75–365 were mapped to strict single-copy genes shared by 20 angiosperm genomes (*De Smet et al., 2013*) (for method of the mapping analysis, see Methods).

The $Ka/Ks$ ratios of the POGs exhibit high similarity in patterns of variation among the 20 taxon pairs. In all pairs, the ratios at peak frequencies are highly similar and more than 90% of the genes are overall under purifying selection (referred to as purifying selection in short) based on their average $Ka/Ks$ ratios < 0.9 across all sites within the genes. The exception was *Nelumbo*, which has 87.6% of the genes with $Ka/Ks$ ratios < 0.9. The greatest proportion of genes under purifying selection was observed in *Gelsemium*, which had 94.9% of the genes with $Ka/Ks$ ratios < 0.9. A very small proportion, from 1.6% (in *Acorus*) to 3.7% (in *Nelumbo*), is evolving nearly neutrally with $Ka/Ks$ ratios in the range of 0.9–1.1, and another small proportion, from 3.1% (in *Gelsemium*) to 8.8% (in *Nelumbo*), is under putative positive selection based on values of $Ka/Ks$ ratios greater than 1.1. A very small proportion, from 0.6% (in *Gelsemium*) to 2.0% (in *Nelumbo*), is under putative strong positive selection with $Ka/Ks$ ratios ranging from 2 to 7 and rarely to 17 (*Figure 3*). The proportion of genes under truly positive selection may be smaller the numbers reported here due to uncertainty in estimation of $Ka/Ks$ ratios when $Ks$ is small.

We arbitrarily divided the genes under purifying selection into three categories, $Ka/Ks < 0.1$ as genes under strong purifying selection, $0.1 < Ka/Ks < 0.5$ as genes under moderate purifying selection, and $0.5 < Ka/Ks < 0.9$ as genes under relaxed purifying selection. Among the taxon pairs, we observed 41% (in *Convallaria*) to 68% (in *Gelsemium*) of the POGs fall into the category of moderate purifying selection, 11% (in *Gelsemium*) to 36% (in *Convallaria*) fall into the category of strong purifying selection, while 9.7% (in *Acorus*) to 19.5% (in *Nelumbo*) fall into the category of relaxed purifying selection. In all taxon pairs, the relative abundance of POGs under the different categories of selection pressures follows the same order, from high to low, as moderate purifying selection, strong

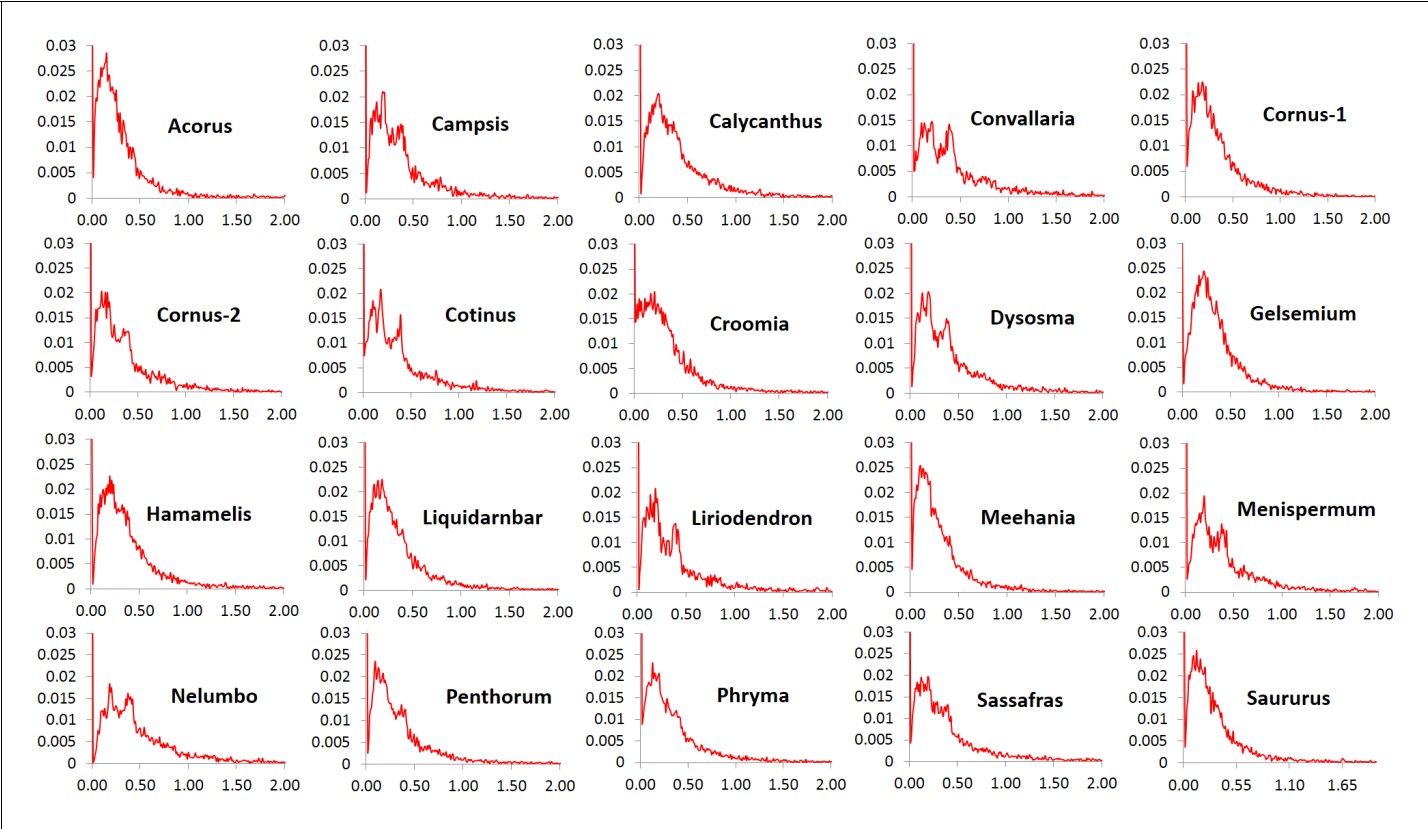

**Figure 3.** Frequency distribution of *Ka/Ks* ratios of putative orthologs (POGs) from leaf transcriptomes of 20 species/variety pairs of angiosperms.

purifying selection, relaxed purifying selection, weak/moderate positive selection, putatively neutral, and strong positive selection (*Figure 4*; *Supplementary file 3*).

To investigate if variation in *Ks* values at peak frequency reflects variation in biogeographic isolation times of taxon pairs, phylogenetic analyses were conducted using the seven POGs shared by all taxa and the 79 POGs shared by ≥90% of the taxa, respectively. The results recovered the relationships of the 20 lineages congruent with the APG IV summary phylogeny and classification (*The Angiosperm Phylogeny Group, 2016*) (*Figure 5*; *Figure 5—figure supplement 1*). Divergence times of the biogeographic pairs estimated using total substitutions of these two sets of gene sequence data and BEAST (*Drummond and Rambaut, 2007*) were similar, with the results from 79 genes without the fossil constraint from *Cornus* slightly younger (*Figure 5*; *Figure 5—figure supplement 1*). The estimated divergence times suggested that isolation of the species pairs occurred in an interval between the late Miocene and Pleistocene (~10.67 —~1.69 mya; *Figure 5*). Species divergence time was positively correlated with the *Ks* value at peak frequency across taxon pairs (*Figure 5*; r = 0.8034; p<0.001; for values used for the correlation analysis, see *Supplementary file 4*). On the other hand, the *Ks* values at peak frequency of the taxon pairs are: (1) positively correlated with the abundance of genes under moderate purifying selection (0.1 < *Ka/Ks* <0.5; r = 0.9324; p<0.001), (2) negatively correlated with the abundance of genes under putative positive selection (*Ka/Ks* >=1.0, r = - 0.5344; p=0.015; 1.1 <= *Ka/Ks* <2, r = - 0.62; p=0.004), and (3) negatively correlated with the abundance of genes under putative strong purifying selection (*Ka/Ks* <0.1; r = - 0.7940; p<0.001) (*Figure 6*; *Supplementary file 3*, *Supplementary file 10*). The divergence times of taxon pairs are also positively correlated with the abundance of genes under moderate purifying selection (0.1 < *Ka/Ks* <0.5; r = 0.73; p<0.001) and negatively correlated with the abundance of genes under strong purifying selection (*Ka/Ks* <0.1; r = - 0.64; p<0.009; *Figure 5*; *Supplementary file 4*). The number of genes under strong positive selection (*Ka/Ks* >2) varies among the taxon pairs from 72 (in *Campsis*) to 184 (in *Nelumbo*) (*Table 2*). In each pair, approximately half of these genes have GO annotations (*Table 2*) that, in combination among all pairs,

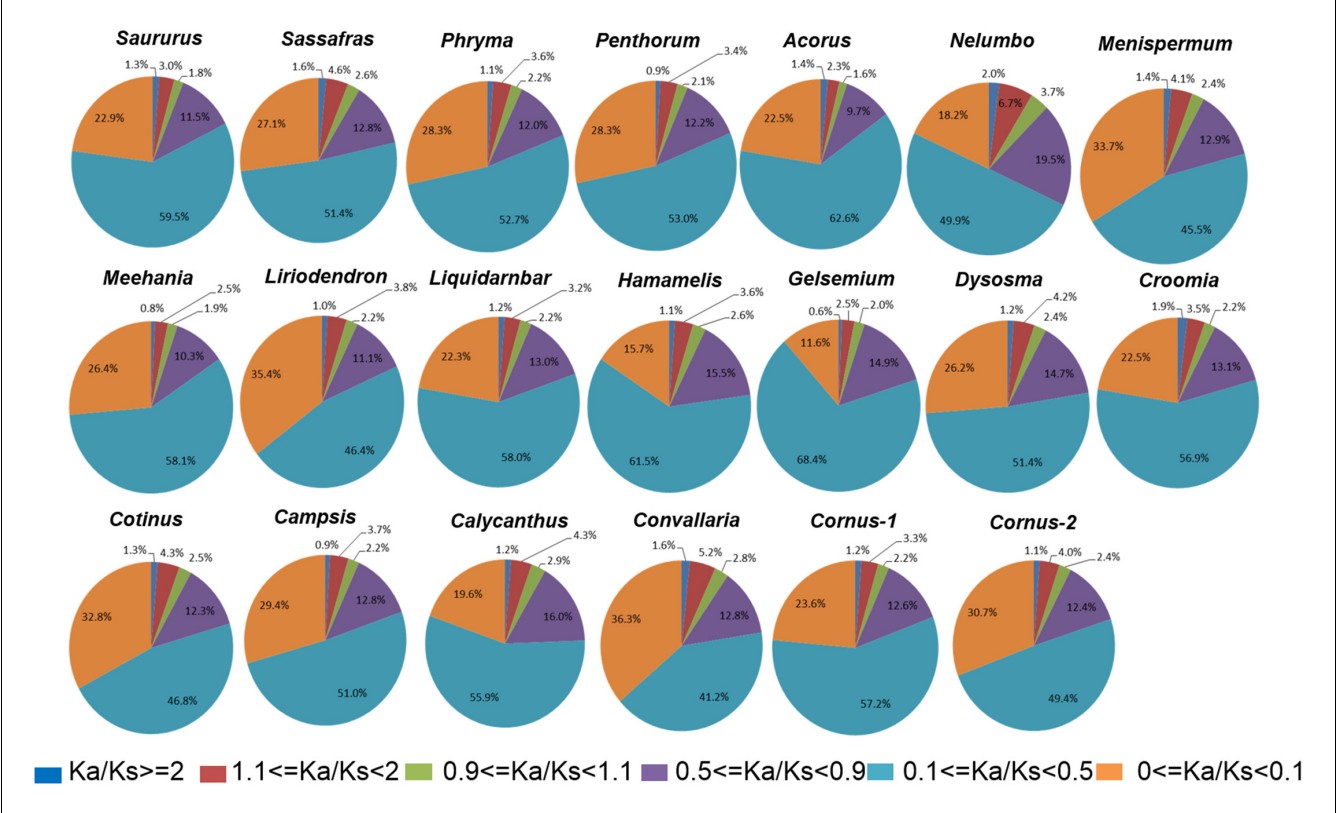

**Figure 4.** Relative abundance of putative orthologs (POGs) with different categories of *Ka/Ks* values. Numbers of genes in each category are provided in *Supplementary file 3*.

mapped to a total of 223 GO terms, varying from 21 (in *Cornus-2*) to 46 (in *Convallaria*) in each taxon pair (*Supplementary file 5*). Many of these GO terms are unique to a single lineage (100/223 = 44.8%).

A majority (59) of these lineage-specific GO terms belongs to the Biological Process (BP) categorization, while 23 belong to the Molecular Function (MF) ontology, and 16 belong to the Cell Components (CC) ontology (*Supplementary file 5*). Fourteen of the BP GO terms were annotated to responses to various stimuli (biotic or abiotic, external or internal stimuli), defense response, immune process, or signal transduction and were distributed among 11 taxon pairs, with each term was present in 1–3 pairs. In all, 124 of the 223 GO terms (55.2%) are shared by two or more of the 20 taxon pairs. Among these, 12 were shared by 10–15 taxon pairs. These 12 genes were annotated to the nucleus, protein complex, hydrolase activity, zinc ion binding, RNA binding, ATP binding, structural constituent of ribosome, oxidoreductase activity, cellular protein modification process, translation, single-organism biosynthetic process, and oxidation-reduction process (*Supplementary file 5*). A single GO term is shared by all 20 taxon pairs and is annotated to integral component of membrane (ICM) in the CC category (*Supplementary file 5*). The specific genes and the relative abundance of genes within each of the CC, BP, and MP categories vary among the taxon pairs (*Figure 7*; *Figure 7—figure supplement 1*; *Figure 7—figure supplement 2*). However, within the CC category, all pairs have the greatest proportion annotated to ICM (*Figure 7*), although they vary in the exact number (*Table 3*). The analysis of the *Ka/Ks* > 2 genes using the variation cluster method (*Wagner, 2007*) showed most of them (>57% in all taxon pairs, >70% in 15 pairs) were also under positive selection (indicated by significant *p* value for test of clustered nucleotide substitutions) (*Supplementary file 6*).

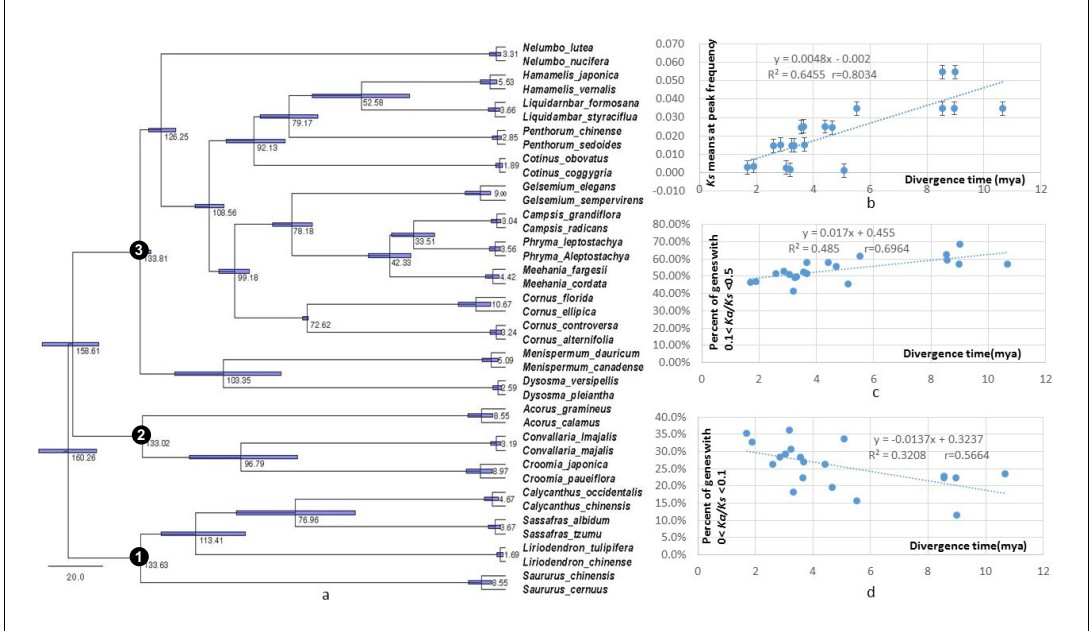

**Figure 5.** Dated global phylogeny of the 20 taxon pairs and correlations of divergence time with level of divergence at sysnonymous sites, withabundance of genes under moderate purifying selection (Ka/Ks = 0.1 - 0.5), and with abundance of genes under strong purifying selection (ka/Ks <0.1). (a) Dated global phylogeny and divergence times of taxon pairs estimated using BEAST program and seven single-copy orthologs shared by all 40 taxa with no missing data. Number 1, 2, and 3 indicate the crown node of Magnoliidae, crown node of Monocotyledoneae, and crown node of Eudicotyledoneae, respectively. (b – d). Correlation between divergence time and (b) $Ks$ value at peak frequency of each genus, (c) abundance of POGs with $Ka/Ks$ values between 0.1–0.5, and (d) abundance of POGs with $Ka/Ks$ values < 0.1. Data used for the analyses are available in ***Supplementary file 4***.

The online version of this article includes the following figure supplement(s) for figure 5:

**Figure supplement 1.** Divergence times of 20 taxon pairs estimated with 79 single copy orthologs present in 90% or more species using BEAST.

## Discussion

### Conserved genomic pattern of evolutionary divergence following geographic speciation

Our results show a general trend of genome-wide molecular divergence in POGs among the phylogenetically divergent genera showing the EA-ENA floristic disjunction or other types of allopatric divergence. That is, similar frequency distributions of $Ks$, $Ka$, and $Ka/Ks$ ratios were observed for the POGs expressed in leaves, with peaks clustered within a narrow range of small values ($Ks$: 0.001– 0.05; $Ka$: 0.002–0.004; $Ka/Ks$: 0.15–0.25) and a long tail of larger values ($Ks$ > 0.075, $Ka$ >0.035, $Ka/Ks$ > 0.5; ***Figure 1***, ***Figure 2***, ***Figure 3***). The pattern we observed indicates that in each of the taxon pair, including the 16 EA-ENA pairs, most of the genes show shallow divergence or evolve slowly at a similar rate, while a small proportion of the genes evolve faster. This suggests that the molecular divergence in the POGs is generally low after geographic speciation. The peak frequency range of $Ks$, $Ka$, and $Ka/Ks$ values revealed substantial differences in the magnitudes among genera, indicating that the taxon pairs have diverged to different extents, likely reflecting their relative timing of divergence and/or varying selection pressures among different genera. Evidence from divergence time estimation supported this prediction as discussed below.

Divergence time estimation using the global phylogeny with DNA sequences of the seven POGs shared by all of the studied species demonstrates variation in times of divergence among genera. Biogeographic pairs with similar $Ks$ values of peak frequency diverged at similar geological times (***Figure 5***). The divergence times of taxon pairs fall into the window of the mid-Miocene to the Pleistocene, largely similar to previously reported times of divergence for these taxa (e.g., ***Xiang et al., 2000***; see summary in ***Wen, 2001***; ***Milne and Abbott, 2002***; ***Ian Milne, 2006***; ***Wen et al., 2010***). Although the dates for some pairs were previously estimated to be older than obtained here (e.g.,

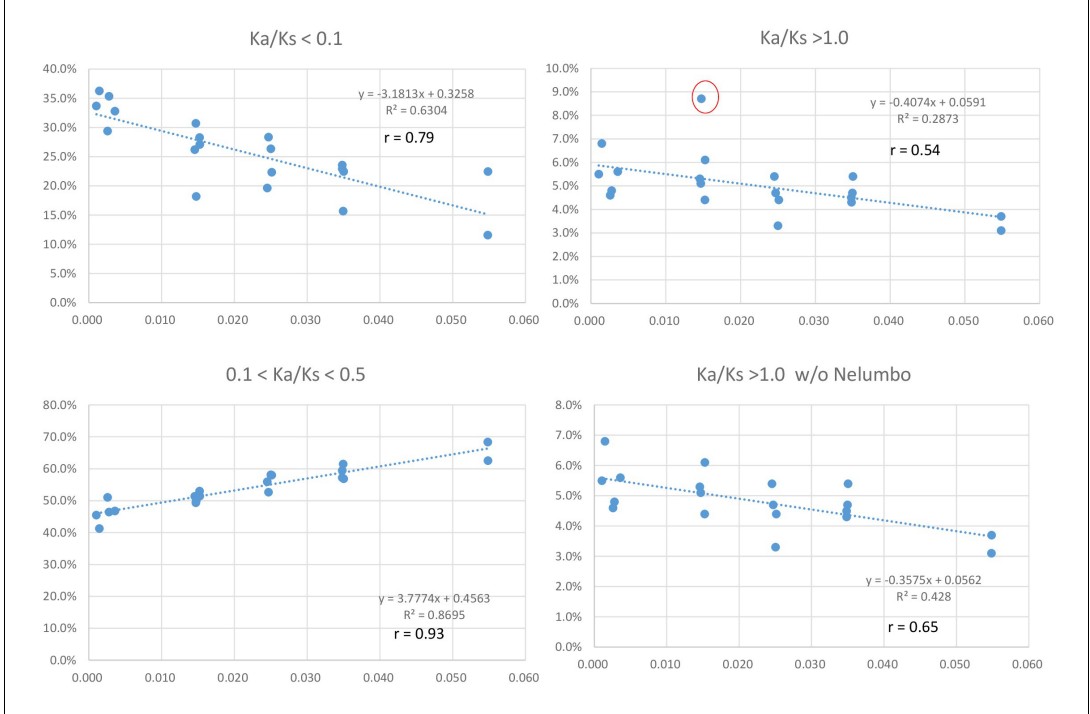

**Figure 6.** Positive and negative relationships between relative abundance (Y-axis) of genes with *Ka/Ks* values in the indicated ranges and *Ks* values of peak abundance (X-axis) in the 20 taxon pairs. The pattern remains when the modified *Ks* (see Materials and methods) is used. Data used for the analyses are available in ***Supplementary file 4***.

*Cornus-1, Cornus-2, Liriodendron, Liquidambar,* and *Sassafras*), those dates also fall in the Miocene (***Wen et al., 2010***). Divergence times similar to those estimated in this study have also been reported for *Cornus* and *Liriodendron* (***Xiang et al., 2000***). The major pattern of divergence revealed here by analyses of many nuclear genes for the 16 EA-ENA disjunct taxon pairs is consistent with the previous mega-synthesis of studies of individual lineages that were largely based on ITS and/or a few plastid gene sequences or non-coding markers (***Donoghue and Smith, 2004***; ***Wen et al., 2010***; ***Harris et al., 2013***). Both showed a major pattern of the Miocene or more recent disjunction of the EA and ENA sister species/varieties.

The divergence times estimated for the EA-NA (ENA, WNA, or NA) disjunct pairs in this study support the importance of a geographical connection through the Bering Land Bridge (BLB) and per-haps also long-distance dispersal (LDD) to explain a part of the EA-ENA floristic similarity. Geological evidence indicates that the BLB was available as a floristic connection throughout most of the Paleogene and Neogene and closed for the potential migration of terrestrial organisms between 7.4 and 4.8 mya (***Marincovich and Gladenkov, 1999***; ***Graham, 2018***), while the North Atlantic Land Bridge (NALB) broke up in the early Eocene (***Marincovich et al., 1990***; ***Tiffney, 2000***).

The disjunction of the EA-NA taxon pairs may have occurred through vicariance (the break-up of what was once a continuous distribution by different climatic cooling or geological events occurring during the Neogene) or founder event (which could have occurred across the BLB to generate a new distribution or could have been a rare LDD event). Gene flow could stop around the time of migra-tion simply because of geographic distance. Even though the BLB might still have been available, pollen and/or seed flow might have been limited given the geographic distance and high latitude of BLB (long, cold winter with short days). Divergence then could have occurred due to either selection or drift if the founder populations were small. On the other hand, if local populations were large and generation times were long, populations would not diverge very quickly under even limited gene flow and a physical break in the distribution. In such cases, divergences might post-date the end of the BLB and other barriers, for example, if disjunctions were formed by vicariance. At present, our data do not permit distinguishing between the alternative hypotheses of vicariance and LDD in

**Table 2.** The number of POGs under strong positive selection ($Ka/Ks > 2$) in each taxon pair and results from Blast Search.

| Genera | Total | With blast (without hits) | With blast hits | With mapping | With GO annotation |
|---|---|---|---|---|---|
| *Acorus* | 101 | 18 | 20 | 17 | 46 |
| *Calycanthus* | 115 | 14 | 20 | 25 | 56 |
| *Campsis* | 72 | 5 | 15 | 10 | 42 |
| *Convallaria* | 123 | 51 | 12 | 15 | 45 |
| *Cornus-1* | 123 | 17 | 16 | 23 | 67 |
| *Cornus-2* | 116 | 7 | 12 | 29 | 68 |
| *Cotinus* | 124 | 40 | 11 | 13 | 60 |
| *Croomia* | 170 | 32 | 29 | 20 | 89 |
| *Dysosma* | 108 | 36 | 14 | 18 | 40 |
| *Gelsemium* | 67 | 5 | 10 | 8 | 44 |
| *Hamamelis* | 114 | 28 | 15 | 26 | 45 |
| *Liquidambar* | 122 | 23 | 19 | 21 | 59 |
| *Liriodendron* | 78 | 15 | 7 | 19 | 37 |
| *Meehania* | 85 | 12 | 14 | 7 | 52 |
| *Menispermum* | 106 | 18 | 6 | 19 | 63 |
| *Nelumbo* | 184 | 9 | 32 | 39 | 104 |
| *Penthorum* | 93 | 5 | 16 | 9 | 63 |
| *Phryma* | 119 | 4 | 24 | 16 | 75 |
| *Sassafras* | 159 | 24 | 27 | 31 | 77 |
| *Saururus* | 102 | 7 | 21 | 16 | 58 |

establishing the EA-NA disjunction in these taxa. However, the fact that 65 genera of seed plants exhibit this disjunction and the availability of NALB and BLB for migration argue for vicariant origins of these taxa and the similar flora; LDD for this number of taxa representing a range of life histories does not seem parsimonious, and may only play a minor role in formation of the similar flora in EA and ENA. Development of phylogeny-based methodology that models speciation under the different scenarios is needed for testing these hypotheses.

Our analyses also revealed a repeated pattern of divergence of POGs among the taxon pairs in allopatric speciation. That is, we observed the same order of relative abundance of POGs under various selection pressures (*Figure 4*), disregarding the relative timing of species isolation (i.e., differences in *Ks* value of peak abundance and divergence time), including those conspecific pairs. Specifically, we found a tiny portion of the POGs are under strong positive selection or putatively evolve neutrally, while most of the POGs are under purifying selection pressures. Although the *Ka/Ks* ratio and variation cluster both have low power for detecting positive selection (leading to underestimation of number of positive selection genes), this pattern is likely true due to functional constraint of protein coding genes. The observed pattern further suggests that most of the molecular evolution (or DNA substitution) in the POGs was neutral (or synonymous). The deciduous forests in eastern Asia and eastern North America are situated at similar latitudes, and both China and the United States have an eastern coast line; thus, they may have similar regional climatic and ecological conditions (*E. Ricklefs et al., 2004*). Therefore, there are presumably low divergence pressures for gene functional evolution in the two areas. One would therefore expect most genes to evolve slowly under purifying selection to conserve ancestral functions and the observed excess synonymous substitutions (in comparison to non-synonymou substitutions) in these genes were results of elimination of nonsynonymous substitutions by natural selection while the neutral synonymous substitutions were fixed by genetic drift. On the other hand, although eastern Asia and eastern North America share similarities in general climate, there must be differences in micro-abiotic and biotic environments between the two regions which could have driven diversifying evolution of some genes in the sister lineages for local adaptation, speciation, and subsequent divergence.

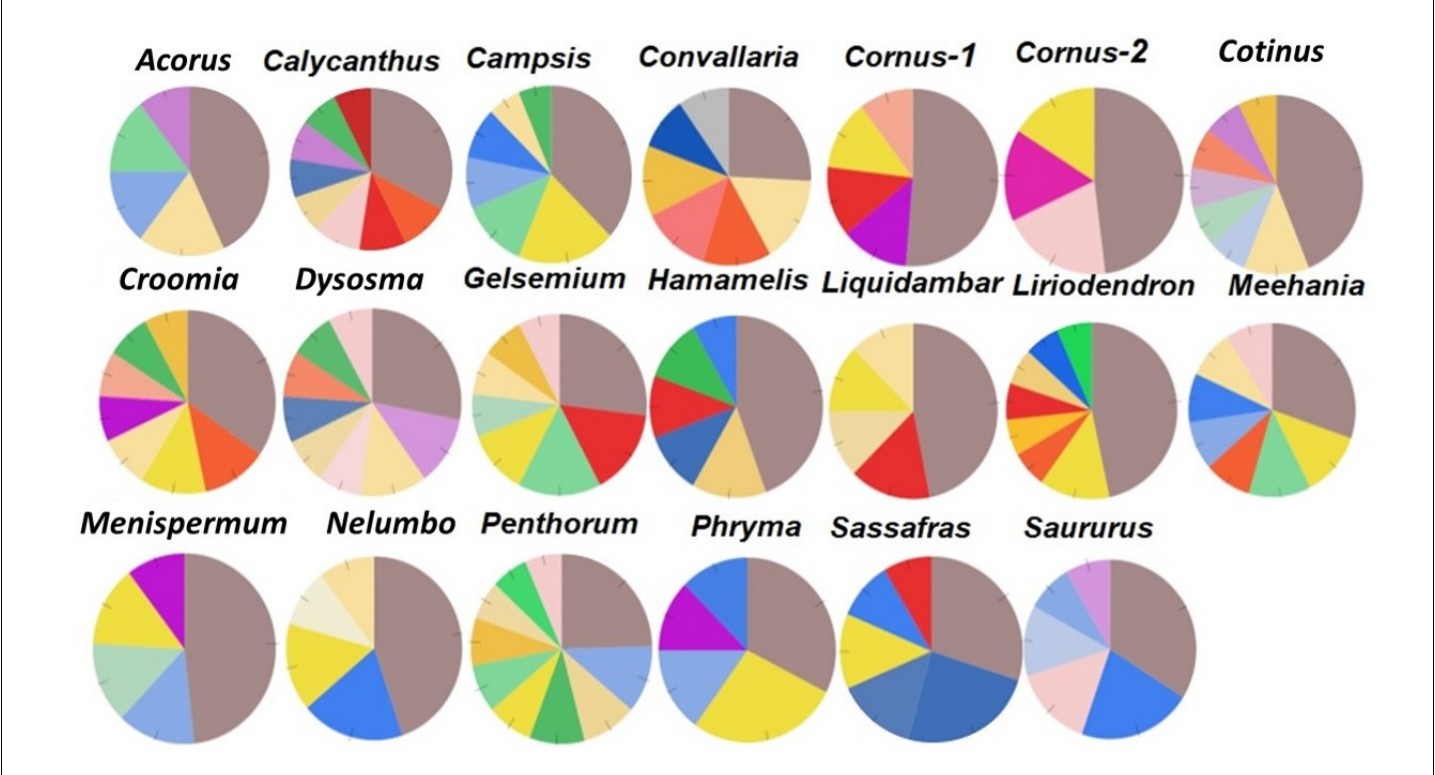

**Figure 7.** Relative abundance of genes with *Ka/Ks* > 2 annotated to Cell Component category, drawn from data in *Supplementary file 5*. Genes annotated to integral component of membrane (ICM) are shown in dark brown.
The online version of this article includes the following figure supplement(s) for figure 7:

**Figure supplement 1.** Relative abundance of *Ka/Ks* > 2 genes annotated to different functions in the Biological Process category; drawn from data in *Supplementary file 5*.

**Figure supplement 2.** Relative abundance of *Ka/Ks* > 2 genes annotated to different functions of in the Molecular Function category; drawn from data in *Supplementary file 5*.

Our results are consistent with these predictions. A dominant majority of the POGs examined in this study were under strong purifying selection pressures and only a tiny portion of the genes (0.6%–2%) were potentially under 'strong' positive selection (*Ka/Ks* > 2) (*Figure 4*). Annotation of these genes evolving under strong positive selection in the EA-ENA taxon pairs pointed to potential abiotic and biotic agents of selection (*Supplementary file 5*; also see discussion below). The observation of this same pattern in non-EA-ENA taxon pairs suggests that this 'conservatism' pattern of gene evolution may be a rule for allopatric speciation of any organism, which allows conservation of the ancestral system as well as opportunities for modification of the ancestral system during speciation. The conserved order of gene abundance, as observed in all taxon pairs in this study, may be essential to maintaining a balance between the ability to preserve the ancestral biological system and the ability to evolve new features that are beneficial for new adaptation. Although genome-wide conservation of genes may have been predicted in the past, we present here empirical evidence from thousandths of putative orthologous genes supporting this evolutionary pattern of conservatism in geographic speciation across a wide array of angiosperm lineages. The repeated pattern of variation of relative abundance of POGs under different selection pressures among taxa may represent a genomic 'rule' that can be tested by examining all gene families in the transcriptomes or genomes of the studied taxa. It is allied with a hypothesis that molecular evolution in the allopatric taxa is mostly neutral involving mostly silent nucleotide substitutions in the genome. This hypothesis may be further tested by examining molecular evolution of genes in additional plant lineages to determine if it is supported and is a rule. Study of three pairs of sunflower species that diverged along latitudinal or longitudinal gradients similarly showed a repeated pattern of genomic

**Table 3.** Number of genes with $Ka/Ks > 2$ ratios annotated to Integral Component of Membrane.

| Genera | Integral component of membrane |
| --- | --- |
| Acorus | 12 |
| Calycanthus | 13 |
| Campsis | 12 |
| Convallaria | 8 |
| Cornus-1 | 20 |
| Cornus-2 | 12 |
| Cotinus | 18 |
| Croomia | 26 |
| Dysosma | 7 |
| Gelsemium | 7 |
| Hamamelis | 16 |
| Liquidarnbar | 15 |
| Liriodendron | 7 |
| Meehania | 10 |
| Menispermum | 14 |
| Nelumbo | 26 |
| Penthorum | 15 |
| Phryma | 13 |
| Sassafras | 21 |
| Saururus | 16 |

divergence (*Renaut et al., 2014*), with evidence implying that both purifying and divergent selection contributed to repeatable patterns of divergence and that repeated genome evolution may have resulted from both similar selective pressures and shared local genomic landscapes.

Our observation of the length of time of species isolation is positively correlated with the abundance of genes under moderate pressure of purifying selection ($0.1 < Ka/Ks < 0.5$), but negatively correlated with the abundance of genes under strong pressure of purifying selection ($Ka/Ks < 0.1$) (*Figure 5*; *Figure 6*), suggests that as geographic isolation or divergence proceeded in time (within the time window of divergence of the species pairs studied), the purifying selection pressures of some genes may have shifted from high intensity during early geographic isolation to lower intensity at the later stage of isolation. The shift might be attributed to changes of evolutionary trajectories of the gene at some sites where replacement substitutions are favored for modification of the ancestral functions leading to relevant morphological/ecological divergence between the species pairs. Alternatively, a relaxed purifying selection intensity of some genes could be simply a result of accumulation of slightly deleterious nonsynonymous substitutions rather than adaptive evolution. The increase of $Ka/Ks$ ratios with time observed in our study is unlikely an artifact of saturation at silent sites of these genes because saturation of silent sites may be expected in comparisons of deep phylogenetic lineages (e.g., family or higher ranks) and unlikely to occur in species/variety pairs. Time-dependent shift of purifying selection intensity have also been reported in study of yeast (*Elyashiv et al., 2010*). The model developed for evolution of purifying selection in this study revealed that most, but not all changes, could be explained by systematic shifts in the efficacy of selection. An observation of a weak negative relationship between $Ks$ peak values and abundance of genes under positive selection pressure ($Ka/Ks > 1$; *Figure 6*) in our study also suggests that as time of isolation proceeded, pressure of positive selection for some genes might have also relaxed to a lower $Ka/Ks$ values due to accumulation of mutations at synonymous sites in the genes. Study in mammal genomes also indicated that positive selection is often episodic (*Kosiol et al., 2008*).

Unfortunately, our study cannot track the evolutionary fate of specific genes within a taxon pair through time to provide direct evidence for these explanations on the observed relationships.

Nonetheless, we examined *Ka/Ks* values of the 79 POGs shared by 90% or more taxon pairs that diverged at different times to see if the same orthologous genes exhibit a relationship between variation of *Ka/Ks* values and time. A simple scattered plot in excel for the divergence times and the *Ka/Ks* values (with *Ka/Ks* values = 0, 99 removed) of the 79 POGs shared by 90% or more taxa revealed that eight of these genes had a correlation at *r* > 0.5, among which two correlations were positive and six were negative (*Supplementary file 7*). The *Ka/Ks* ratios of POGs that showed a correlation with times were all smaller than 1.0 (all <0.5 except for a couple of cases). These data provide another line of indirect evidence congruent with the hypothesis that some genes may have indeed shifted the pressure of purifying selection (intensified or relaxed) during the evolutionary divergence of the taxon pair. When only including *Ka/Ks* ratios smaller than 0.5 in the analysis of these 79 POGs, the number of genes showing correlation with time increased (9 positive and seven negative relationships with *r* > 0.5; *Supplementary file 8*). The relationships are strong in a few genes for either pattern. This observation suggests, perhaps, the presence and elimination of slightly deleterious nonsynonymous substitutions play a major role in the changes of *Ka/Ks* ratios. Lack of correlation between *Ka/Ks* ratios and time for most of the 79 POGs also suggests that the evolutionary fate of most of these POGs is more taxon-specific independent of time. Although most of them were under purifying selection, a handful cases were under putative positive selection with (*Ka/Ks* > 1; *Supplementary file 7*) which are more found in the aquatic *Nelumbo* and *Saururus* genera.

## Genes under strong positive selection

The number and predicted functions of genes under strong positive selection (*Ka/Ks* > 2) varied among the disjunct pairs, and a large proportion of the GO terms are unique to a single genus, suggesting that the divergence and evolutionary adaptation of the species in each genus has been likely facilitated by different molecular mechanisms and biological processes. For example, among the 20 lineages we examined, the aquatic eudicot *Nelumbo* has the largest proportion of its POGs under positive selection and the smallest proportion of POGs under purifying selection, while the reverse is found in the terrestrial twining vine *Gelsemium*, a member of the asterid clade Gentianales. The two species of *Nelumbo*, *N. nucifera* (EA) and *N. lutea* (ENA), differ in four features of their reproductive structures, including tepal color and persistence, receptacle color, and fruit shape. Gene annotation identified 12 genes under strong positive selection for response to stimulus, nine for regulation of gene expression, seven for developmental process, eight for DNA-templated transcription, and eight for cellular lipid metabolic process. These genes may be interesting for future investigation into their roles in observed morphological divergence (*Supplementary file 5*). In *Gelsemium*, 12 morphological characters in reproductive structures (including inflorescence size, floral shape/size, fruit and seed morphology, etc.; see list in *Wyatt et al., 1993*) and two vegetative structures (petiole length and leaf shape) differ between the two disjunct species (*Wyatt et al., 1993*). Gene annotation for strong positive selection genes in this genus identified two for plastid organization, two for photosynthesis (light reaction), three for response to stress, two for establishment of localization in the cell, three for oxidation-reduction process, two for response to light stimulus, three for proteolysis, two for regulation of biological quality, two for negative regulation of metabolic process, three for signal transduction, and two for RNA processing, among others. The biological processes involving the positively selected genes all differ between the two genera except for DNA metabolic process, which was common to both, involving seven genes in *Nelumbo* and two genes in *Gelsemium* (*Supplementary file 5*).

It is difficult to decipher if there is any link between the positively selected genes and morphology without additional evidence, but these genes may serve as candidates for further detailed population genetics and evo-devo studies to test their roles. Many of the positively selected genes could also have been involved in divergence of cryptic characteristics that are unknown, and thus serve as a guide for future investigation. The GO terms of these genes (*Supplementary file 5*) support that possible ecological/environmental pressures or selection agents that differ between the growing habitats of the taxon pairs in EA and ENA or other areas may have driven the extent of divergence in these genera. For example, the positive selection genes (*Ka/Ks* > 2) with GO terms in our study are enriched for immunity/stimuli responses. Similar case is also found in positive selection genes in mammals which were enriched for a wide variety of functions related to immunity and defense (*Kosiol et al., 2008*). Identification of the specific sites under positive selection in these genes would be helpful to guide future studies on the potential functional consequences of selection.

Unfortunately, our comparisons of two sequences for each gene in each taxon pair did not have the power for such analysis. Further analyses of these genes with increased taxon/population sampling will be needed to identify the particular sites of selection.

The single GO term common to all 20 taxon pairs was annotated to integral component of membrane (ICM) and was most abundant among the cell component terms (*Figure 7*; *Supplementary file 5*, row in bright red). This also suggests that a potential common regional force acting on these taxa resulted in the adaptive evolution of ICM genes, which may be involved in some of the common biological processes, such as signaling associated with host-microbe interactions and responses to various stimuli. When examining the names of these integral cell membrane component genes, they seem to vary among taxa, and it is difficult to determine if they are involved in similar biological processes in different taxa, without additional evidence (*Supplementary file 9*). These positively selected genes of integral cell membrane component provide candidates for future studies to understand the causes and consequences of their molecular evolution.

Recent studies have revealed repeated patterns of genomic divergence associated with species formation (*Pereira et al., 2016*). Such patterns was considered as evidence suggesting that natural selection tends to target a set of available genes (*Pereira et al., 2016*). Our study found a set of genes was targeted for strong positive selection in each taxon pair, suggesting that these genes may play a role for the allopatric divergence of the studied taxon pairs. Although the set of genes differed among taxon pairs, 12 were shared by 10–15 pairs and one of them was share by all pairs. Whether these common targets reflect shared biological processes unique to the allopatric divergence of the studied taxa or general to allopatric or even sympatric divergence of all taxa need to be tested by further studies.

## Uncertainty in gene copy, orthology, Ka/Ks estimation, and divergence time estimation

We stress that the POGs in leaf transcriptomes analyzed here are putative orthologs between the two species of each biogeographic pair found in the transcriptomes by OrthoMCL followed by analyses using customized scripts (see Methods below). As discussed above, some of them may represent paralogous copies of the low-copy gene family, but such cases are expected to be uncommon due to the close relationship between the compared taxa of each pair. In these taxon pairs, conservation of expression of a gene copy in the same tissue of the same growth stage is likely. Rare cases of non-homology of POGs would not affect the conclusions derived from thousands of putatively orthologous comparisons. OrthoMCL is known to perform reasonably well in identifying orthologous genes (*Chen et al., 2007*; *Altenhoff and Dessimoz, 2009*). Unfortunately, we do not have genome sequences for any pairs of taxon to confirm the orthology via phylogenetic analyses of gene families. We conducted phylogenetic analyses of the seven POGs shared among the 20 taxon pairs and 79 POGs shared by >90% taxon pairs, respectively, using raxml. The results showed all taxon pairs were grouped as sister taxa in each of the gene trees, in accordance with the expectation that the sequence pairs are orthologous genes. Otherwise, we would have observed separation of sister taxa on the gene tree.

We are aware that uncertainty in $Ka/Ks$ ratios is high when $Ks$ is zero or close to zero. This uncertainty may mislead the identification of genes under strong positive selection as discussed above. In our data, most of the genes (except a few) with $Ka/Ks > 2$ had $Ks > 0.005$ (*Supplementary file 12*). Our plots of the $Ks$ vs. $Ka/Ks$ values in each taxon pair show that the genes with $Ka/Ks > 2$ have a range of $Ks$ values from small to large in all taxon pairs (*Appendix 1—figures 3–22*), indicating the candidate genes under positive selection detected in the study are unlikely artifacts of small $Ks$.

We also recognize that the divergence times of each taxon pair calculated using BEAST (*Drummond and Rambaut, 2007*) and the global phylogeny including all 20 pairs may be somewhat underestimated due to sparse sampling of a deep phylogeny containing both very long and short branches and few calibrations only at deep nodes (see Methods and *Xiang et al., 2011*) that may fall beyond the smoothing capability of BEAST. However, the results are very similar to previous estimates based on *rbcL* sequences (*Xiang et al., 2000*). Despite the possible bias, the influence may be similar across the tree and does not renter their suitability for correlation analyses conducted in the study because the relative order of divergence of the species pairs estimated here is expected to be consistent. Although the divergence times may have been underestimated, we do not think it will not render the implication of a major pattern of BLB migration for the studied taxa because the

divergence time need to be ~20–40 million years older for migration across the North Atlantic land bridge.

## Conclusions

Our comparative analyses of putative single-copy or low-copy genes from transcriptome data of 20 angiosperm lineages in Eurasia and North America reveal a consistent pattern of molecular divergence following allopatry, supporting a potential universal conservatism rule of genomic architecture governing the evolution of these genes. Our data of temporal isolation in the Neogene (or more recent) of the 16 EA-ENA taxon pairs also support an important role of the Bering Land Bridge for intercontinental migration of these taxa. We also identified a total of over 200 genes with annotation under putative strong positive selection following allopatric speciation across these genera, with some annotated to biological processes responding to various stimuli, providing a pool of candidates for future studies to understand the link between speciation and molecular and morphological divergence.

## Materials and methods

### Sampling, RNA extraction, and transcriptome sequencing

Leaf materials were collected from wild or cultivated plants grown in botanical gardens or arboreta (*Supplementary file 11*) and flash frozen in liquid $N_2$ and stored at −80˚C until RNA extraction was conducted. Total RNA was then extracted from this tissue using the CTAB method of Jordon-Thaden et al. (*Jordon-Thaden et al., 2015*) (protocol number 2) with the addition of 20% sarkosyl. DNA was removed using a Turbo DNA-free kit (Invitrogen, Carlsbad, CA, USA). Extraction success was measured using Bioanalyzer metrics for the quality and quantity of RNA isolated (Agilent Technologies, Santa Clara, California, USA). The pure RNA was sent to BGI (Shenzhen, China) for library construction and sequencing after drying in specially coated tubes (i.e., GenVault, now renamed as GenTegra; IntegenX, Pleasanton, California, USA) that inhibit RNase activity and stabilize the RNA at room temperature. Ribosomal RNAs were removed from the total RNA by the Ribo-Zero rRNA removal kit for plant leaves (Epicentre, Madison, WI, USA) before cDNA library construction. Non-normalized mRNA libraries with insert sizes of ~200 bp were sequenced (paired-end, 100 bp) using the Illumina HiSeq 2000 platform to yield 4–8 Gb each.

### Sequence assembly and identification of putative orthologous genes

The flowchart of transcriptome data analyses is illustrated in *Appendix 1—figure 2*. For each species, prior to contig assembly, reads with poor quality were deleted from the data set, following the standard protocol of Illumina sequencing. Specifically, raw reads were trimmed at the 3' end when the Phred quality score of a read dropped below Q = 20 (or 0.01 probability of error) for two consecutive bases. All 5' and 3' stretches of ambiguous 'N' nucleotides and sequences of less than 20 bp were removed from sequence trimming using CLC Genomics Workbench 4.6.1 (CLC Bio, Aarhus, Denmark). Reads of each species were assembled into contigs by Trinity software (*Grabherr et al., 2011*). Then, for each taxon pair, POGs were identified using orthoMCL (*Li et al., 2003*) and custom Perl scripts as follows. Generally, default parameters were selected for each step of this gene-selection pipeline. "OrthoMCL starts with reciprocal best hits within each transcriptome/genome as potential in-paralog/recent paralog pairs and reciprocal best hits across any two genomes as potential ortholog pairs. Related proteins are interlinked in a similarity graph. Then MCL (Markov Clustering algorithm,Van Dongen 2000; www.micans.org/mcl) is invoked to split mega-clusters'. The key steps in OrthoMCL included (1) 'orthomclFilterFasta' step: the program removes low-quality sequences with sequence length <30 bp and percent stop codons > 20%; (2) 'blastall' step: the program runs all-v-all Blastn with filtered sequences in the above step with default parameters '-v 100000 -b 100000 m 8 -e 1e-5'; (3) 'orthomclPairs step: the program finds pairs of proteins that are potential orthologs, in-paralogs, or co-orthologs; (4) the 'mcl' program split mega-clusters into the final OrthoMCL ortholog groups; and (5) the ortholog groups file is outputted by OrthoMCL. In the ortholog groups file for a taxon pair, if a gene group contains a single gene from each taxon, this group was then picked as a 'single-copy ortholog group' using a custom Perl script. We refer here to these genes as putative orthologs (i.e. POGs) and note that they are 'single-copy' in the context

of the comparisons – i.e., transcriptomes of a single tissue in a single taxon pair – and thus may not truly be single-copy genes within either of the taxa in each comparison. Each 'single-copy' ortholog of a taxon pair was further analyzed by scanning the ortholog groups file using the protein sequence-based ESTScan software (*Lottaz et al., 2003*) for CDS. The CDS sequences of each single-copy ortholog were aligned and converted into PAML format by MAFFT software (*Katoh and Standley, 2013*) and custom Perl scripts. The Perl scripts and complete pipeline (GDMET – Genome Duplication and Molecular Evolution by Transcriptome Sequences) developed for these data analyses is available at https://github.com/ybdong919/GDMETS/releases. The NCBI Bioproject number and Biosample numbers for the raw transcriptome data are provided in *Supplementary file 11*.

To see how many of these POGs were represented in the 'global' single or low copy gene data sets of seed plants (*Li et al., 2017*); identified from a combination of genome and transcriptome data) and flowering plants (*De Smet et al., 2013*); identified from 20 angiosperm genomes), respectively, we performed the following mapping analyses. First, the single-copy or low-copy gene names of *Arabidopsis thaliana* shared by seed plants or flowering plants were downloaded from the respective publications. Second, the dataset including all cDNA sequences of *Arabidopsis thaliana* was downloaded from its genome database (TAIR10: https://www.arabidopsis.org/). Then the cDNA sequences of the single-copy or low-copy gene of *A. thaliana* shared by seed plants or flowering plants were selected out from the dataset of all cDNA using gene names by Perl script. Finally, the cDNA sequences from each of the 20 taxon-pairs were mapped against the selected gene cDNA sequences of *Arabidopsis thaliana* by Blast program, and those that were mapped by the cDNA from each of the 20 taxon pairs were selected and recorded by Perl script.

## Molecular analyses of POGs – rate of substitution, selection, and gene annotation

To examine the level of molecular divergence and identify genes under positive selection following geographic isolation, synonymous substitutions per site ($Ks$), nonsynonymous substitutions per site ($Ka$), and the ratio ($Ka/Ks$) were calculated for each POGs with estimated standard error (SE) in each taxon pair. The 'yn00' model in the PAML package (*Yang, 1997*; *Yang, 2007*) was used to calculate $Ks$, $Ka$, and their ratio values. A complete pipeline (GDMET) was developed for these data analyses (available at https://github.com/ybdong919/GDMETS/releases). The yn2000 method (*Yang and Nielsen, 2000*) estimated synonymous and nonsynonymous substitution rates with corrections for multiple substitutions at the same site and takes into account of transition/transversion rate bias, base/codon frequency bias.

To test if the level of sequence divergence at synonymous sites observed in most genes is reflective of the relative divergence time of a taxon pair, the relationship between divergence time estimated using genes shared among taxon pairs and the global phylogeny including all taxon pairs (see below for details) and $Ks$ values at the peak frequency in each taxon pair was evaluated using a CORREL function in Excel. The $Ks$ values of POGs in each taxon pair were divided into groups of segments that span 0.01 units (e.g., 0–0.01, 0.01–0.02, 0.02–0.03. . .). The medium value of the $Ks$ range with the highest frequency (referred to as '$Ks$ value at peak frequency') was used for the correlation analyses with medium value of the estimated divergence time. This comparison involved potential overlap of some data (i.e., the synonymous substitutions), because the gene pool with the $Ks$ value that is most frequent in the transcriptome data may include the genes used for divergence time dating. Although the divergence time analysis used total substitutions (including both $Ka$ and $Ks$) in the selected genes (seven shared by all 20 taxon pairs and 79 shared by 90% or more taxon pairs; see details below), we realize that overlapping of the $Ks$ data can reduce the stringency of the analysis, if present. Nonetheless, the analysis can show if the phylogeny and total substitutions-based divergence time is correlated with the level of $Ks$ divergence from pair-wise comparison. If they are correlated, it will suggest that the divergence time estimated with all characters in a small set of genes that are common among taxon pairs can be predicted by the most frequent $Ks$ value from pair-wise comparisons of the genome-wide set of genes in the taxon pair, and both reflect the divergence level of the taxon pair.

To explore the variation pattern of selection pressures of POGs within the genome, we calculated the relative abundance of genes under different ranges of selection pressures. To gain insights into how the variation pattern has changed over time, we examined relationships of the $Ks$ values at peak frequency and divergence time with the abundance of genes under different selection

intensities (i.e., *Ka/Ks* values in different ranges) by correlation analysis using CORREL function in Excel to obtain correlation coefficients. In a few taxon pairs, two consecutive *Ks* ranges have very similar frequencies. We then re-ran the correlation analyses using a modified *Ks* value for comparison, as follows. The modified *Ks* is the middle value of the *Ks* range of the two frequencies if they differ by <10% of the combined frequency, e.g., if *Ks* 0.01–0.02 had a frequency of 21%, *Ks* 0.02–0.03 had a frequency of 19%, the *Ks* in "peak frequency" was determined as 0.02. The difference between 21% and 19% is 2%, less than 10% of 21%+19% (40%). The results using modified *Ks* show no changes in the trends of relationships reported using *Ks* at peak frequency.

The POGs with *Ka/Ks* values greater than two were annotated by Blast2Go v5 (*Conesa et al., 2005*). First, Blastx-fast was used to identify similar, potentially homologous sequences from the green plants (No. taxa: 33,090, Viridiplantae) subset of the non-redundant protein database. A value of 1.0E-5 was assigned to Blast expectation value (E-value), while the other arguments were default. All IDs of these similar sequences were matched against the Gene Ontology (GO) database. Based on the information from Blast and mapping, the most plausible Gene Ontology terms were assigned to the input sequences. We further executed the InterPro step in parallel to the Blast step to identify protein domains and families from the InterPro collection of databases. The functions based on domains and families were merged to the functions from the Blast hits of gene sequences and gene mapping (for details of gene mapping, see introduction of Blast2GO, as described above).

The POGs with *Ka/Ks* values greater than two in each taxon pair were also analyzed with variation cluster method (*Wagner, 2007*) to see how many of these genes also contain significant signal of clustered substitutions, a deviation from neutral molecular evolution and expectation of positive expectation. For each pair of these POG sequences considered, we inferred a codon-preserving local alignment (*Smith and Waterman, 1981*), removed alignment gaps and identified all positions that differed in that alignment using a custom perl script. The coordinates of these differences were used as input to the varclus package (*Wagner, 2007*).

## Phylogeny estimation and divergence times of taxon pairs

The divergence time of each taxon pair was estimated from a phylogeny of all 20 taxon pairs inferred from (1) the seven POGs shared by all 40 samples and (2) those POGs shared by at least 90% of the 40 samples (79 genes) detected by GDMET. These gene sequences were aligned using MAFFT (https://mafft.cbrc.jp/alignment/software/ (*Katoh and Standley, 2013*), visualized and adjusted using Mesquite, and exported in Nexus format (*Maddison and Maddison, 2017*). The best model of molecular evolution (GTR model with a gamma rate parameter) was selected using jModeltest v2.1.6 (*Posada, 2008*), and BEAST v.1.8.2 (*Drummond and Rambaut, 2007*) was used to infer the phylogeny of the 20 genera and estimate the divergence times of each taxon pair (see more details below).

The ages for the crown node of *Magnoliidae*, crown node of *Monocotyledoneae*, and crown node of *Eudicotyledoneae* (see *Figure 5* for these clades) were each constrained as noted below, following the results of previous studies (*Magallón et al., 2015*). Additionally, the crown node of *Cornus* (represented by two pairs from the clades of the earliest split of *Cornus* phylogeny (*Xiang and Thomas, 2008*; *Xiang et al., 2011*) was constrained based on the oldest fossil of the genus from the late Cretaceous (72.1–83.6 mya) for the analysis using seven genes (*Xiang et al., 2011*; *Atkinson et al., 2016*). These constraints were set using uniform priors as the lower and upper bounds. Specifically, the prior for the *Magnoliidae* crown node was set as 127.2–135.2 mya, the prior for the *Monocotyledoneae* crown node was set as 131.0–135.0 mya, the prior for *Eudicotyledoneae* crown node was set as 131.7–135.0 mya, and the prior for the crown node of *Cornus* was set as 72.1–83.6 mya. For the analysis with 79 genes, the *Cornus* node was not constrained to see if results are similar.

The prior parameter settings for the analysis were checked with empty runs without the sequence data first, which resulted in divergence time estimates for the constrained nodes similar to the constrained values with ESS over 200. The distribution models of the other parameters, such as those on molecular evolution, resulting from the empty analyses were then implemented in the real analysis with data. The analysis with real data was run under a GTR model with a gamma rate parameter based on the results of jModelTest, with an uncorrelated lognormal relaxed clock (*Drummond et al., 2006*) and the yule process model (*Stadler, 2010*). The yule.birth rate was set as 0.06, the average rate for plants (*De Vos et al., 2015*). The analysis was run for 10 million

generations, with sampling of trees every 1000 generations. The quality of the runs and parameter convergence were assessed using Tracer v. 1.6.0 (*Rambaut and Drummond, 2009*). The maximum credibility tree of mean heights was then constructed using TreeAnnotator after discarding 1000 trees as burn-in. The tree result was visualized in Figtree v 1.3 (*Rambaut, 2010*).

## Availability of data and materials

The POG sequence pairs and values of *dN, dS*, and *dN/dS* are available at Dryad (https://datadryad.org//). Raw transcriptome data are available at NCBI SRA database with Bioproject number PRJNA508825 and Biosample number from SAMN10534244 to SAMN10534283 (*Supplementary file 11*). Other data sets supporting the conclusions of this article are included in the additional files of the article. Custom analysis scripts are available from GitHub: https://github.com/ybdong919/GDMETS/releases (*Dong, 2018*).

## Acknowledgements

We thank Jeffrey Thorne and Jianhua Li for discussion and critical comments on data analyses and manuscript, Gavin Conant for advice and scripts for running the variation cluster analyses, Andrea Wagner for providing the varclus package, Xiang Liu, Enxiang Li, Jeffery R Hubbard, and Christine Edwards for assistance in collecting leaf samples, Xiang Liu for shipping samples, and the JC Raulston Aboretum, Plant Delights Nursery, and the Sarah P Duke Gardens for providing source plants for sampling. We also thank the anonymous reviewers for their comments for improvement of the manuscript. This work is also supported by the USDA National Institute of Food & Agriculture Hatch Project.

## Additional information

### Funding

| Funder | Grant reference number | Author |
| --- | --- | --- |
| National Science Foundation | DEB-442161 | Yibo Dong<br>Wenbin Zhou<br>Jenny Xiang |
| National Science Foundation | DEB-442280 | Shichao Chen<br>Pamela S Soltis<br>Douglas E Soltis |
| National Science Foundation of China | 31461123001 | Cheng-Xin Fu<br>Yun-peng Zhao |
| National Science Foundation of China | IOS-024629 | Shichao Chen |

The funders had no role in study design, data collection and interpretation, or the decision to submit the work for publication.

### Author contributions

Yibo Dong, Software, Formal analysis, Investigation, Visualization, Methodology, Writing—original draft, Writing—review and editing; Shichao Chen, Formal analysis, Investigation, Methodology, Writing—original draft, Writing—review and editing; Shifeng Cheng, Data curation, Software, Formal analysis, Investigation, Methodology, Writing—review and editing; Wenbin Zhou, Software, Formal analysis, Investigation, Visualization, Writing—review and editing; Qing Ma, Investigation, Writing—review and editing, Collection of experimental materials; Zhiduan Chen, Resources, Investigation, Writing—review and editing; Cheng-Xin Fu, Resources, Investigation, Project administration, Writing—review and editing; Xin Liu, Conceptualization, Resources, Funding acquisition, Investigation, Writing—review and editing; Yun-peng Zhao, Conceptualization, Resources, Funding acquisition, Investigation, Project administration, Writing—review and editing; Pamela S Soltis, Conceptualization, Resources, Supervision, Funding acquisition, Investigation, Methodology, Project administration, Writing—review and editing; Gane Ka-Shu Wong, Conceptualization, Funding acquisition,

Investigation, Writing—review and editing; Douglas E Soltis, Conceptualization, Resources, Data curation, Supervision, Funding acquisition, Investigation, Methodology, Project administration, Writing—review and editing; Qiu-Yun(Jenny) Xiang, Conceptualization, Resources, Data curation, Supervision, Funding acquisition, Investigation, Methodology, Writing—original draft, Project administration, Writing—review and editing

### Author ORCIDs
Shichao Chen https://orcid.org/0000-0002-7160-4245
Qiu-Yun(Jenny) Xiang https://orcid.org/0000-0002-9016-0678

### Decision letter and Author response
Decision letter https://doi.org/10.7554/eLife.45199.sa1
Author response https://doi.org/10.7554/eLife.45199.sa2

## Additional files
### Supplementary files
• Supplementary file 1. Total number of orthologous gene families, number of putative single copy orthologous gene pairs (POGs) in each taxon pair, and the proportion of POGs.

• Supplementary file 2. Frequency of Ks values in 0.01 intervals of all pairs.

• Supplementary file 3. Number and relative abundance in percentage of POGs with $Ka/Ks$ values distributed in different interval categories.

• Supplementary file 4. Information of divergence time estimated from BEAST using seven SCG genes shared by all 40 taxa on a global phylogeny, $Ks$ value in peak frequency, and relative abundance of POGs in three Ka/Ks ratio ranges used in correlation analyses shown in *Figures 5* and *6*.

• Supplementary file 5. Results of gene annotation for genes with $Ka/Ks > 2$.

• Supplementary file 6. Results from Variation Cluster analysis of POGs with $Ka/Ks > 2$.

• Supplementary file 7. Data and results from scatter plot between divergence time and $Ka/Ks$ ratio of 79 POGs shared by 90% or more taxon pairs.

• Supplementary file 8. Data and results for scatter plot of divergence time and $Ka/Ks$ ratio (with $Ka/Ks > 0.5$ data points removed) of 79 POGs shared by 90% or more taxon pairs.

• Supplementary file 9. Functions of genes annotated as integral_compotent_of_membrane predicted from Blast2Go analysis.

• Supplementary file 10. Relationships among $Ks$ in peak abundance, relative abundance of genes in different ranges of selection pressures ($Ka/Ks$ ratios), and divergence time.

• Supplementary file 11. Source of leaf materials used for the study.

• Supplementary file 12. Number of genes with different category value of Ks and Ka/Ks in each species pair.

• Transparent reporting form

### Data availability
Sequences of orthologous gene families and pairs of POGs sequences (prior to EST Scan) used for calculation of Ka and Ks have been submitted to Dryad (https://doi.org/10.5061/dryad.f1f0q44). Raw transcriptome data have been submitted to NCBI SRA database with BioProject number PRJNA508825 and BioSample number from SAMN10534244 to SAMN10534283 (Supplementary file 11).

The following datasets were generated:

| Author(s) | Year | Dataset title | Dataset URL | Database and Identifier |
|---|---|---|---|---|
| Dong YB, Chen SC, Cheng SF, Zhou WB, Ma Q, Chen | 2019 | Natural selection and repeated patterns of molecular evolution following allopatric divergence | https://www.ncbi.nlm.nih.gov/bioproject/?term=PRJNA508825 | NCBI BioProject, PRJNA508825 |

| ZD, Fu CX, Liu X, Zhao YP, Soltis PS, Soltis DE, Xiang QY | | | | |
| Dong Y, Chen S, Cheng S, Zhou W, Ma Q, Chen Z, Fu C, Liu X, Zhao Y, Soltis PS, Wong GK, Soltis DE, Xiang Q | 2019 | Data from: Natural selection and repeated genome-wide patterns of molecular evolution following allopatric divergence | https://doi.org/10.5061/dryad.f1f0q44 | Dryad Digital Repository, 10.5061/dryad.f1f0q44 |

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

## Appendix 1

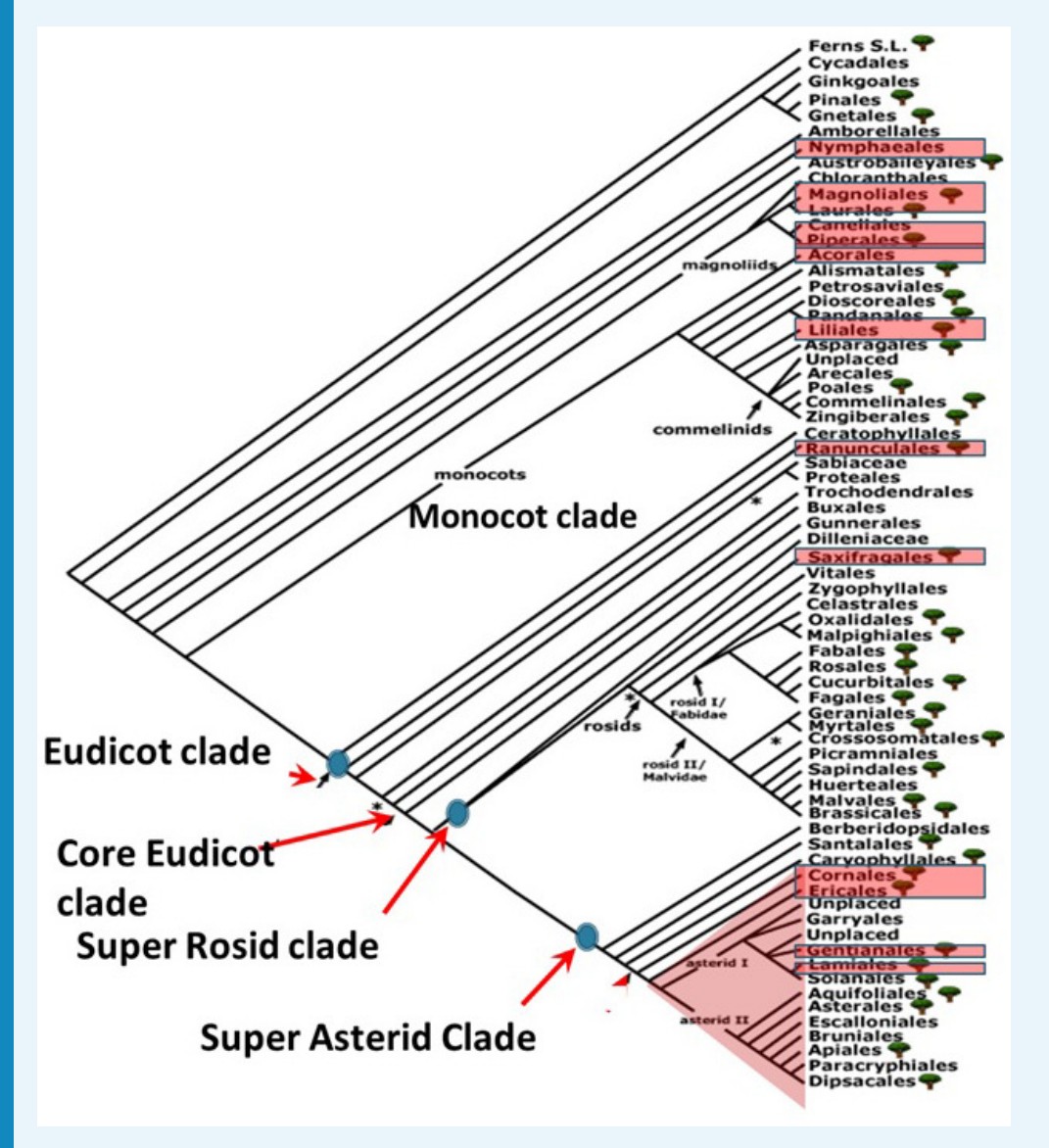

**Appendix 1—figure 1.** Phylogenetic positions of orders represented by species pairs sampled in the angiosperm phylogeny. Phylogenetic tree was taken from Angiosperm phylogeny website.

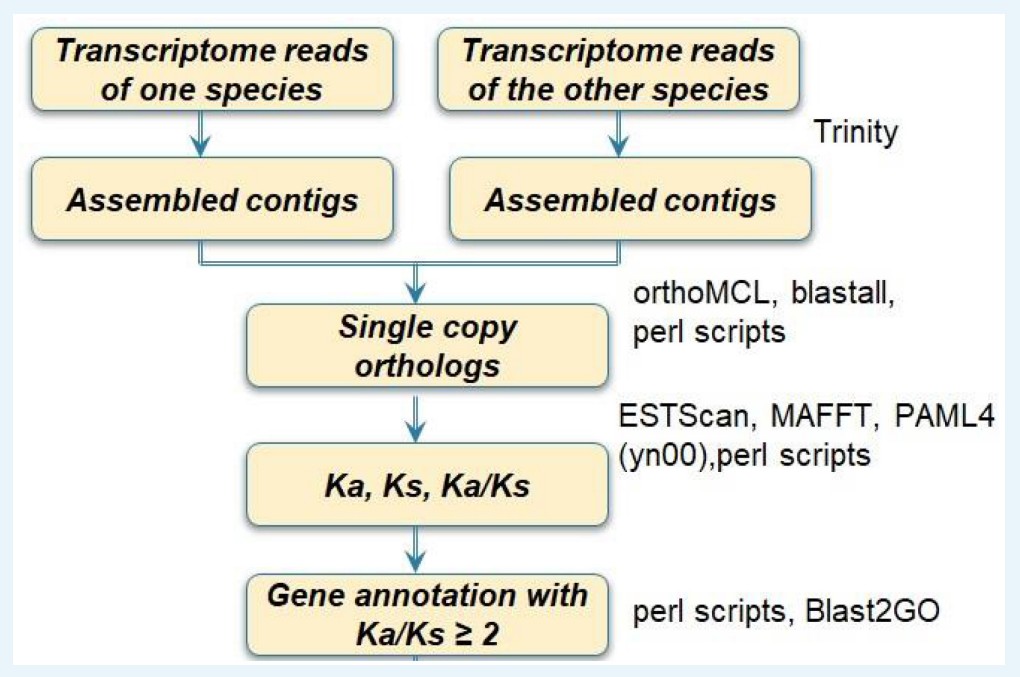

**Appendix 1—figure 2.** Flowchart of transcriptome sequence analyses for one species pair. The process was repeated for 20 pairs. A customized pipeline of programs was developed for running these steps.

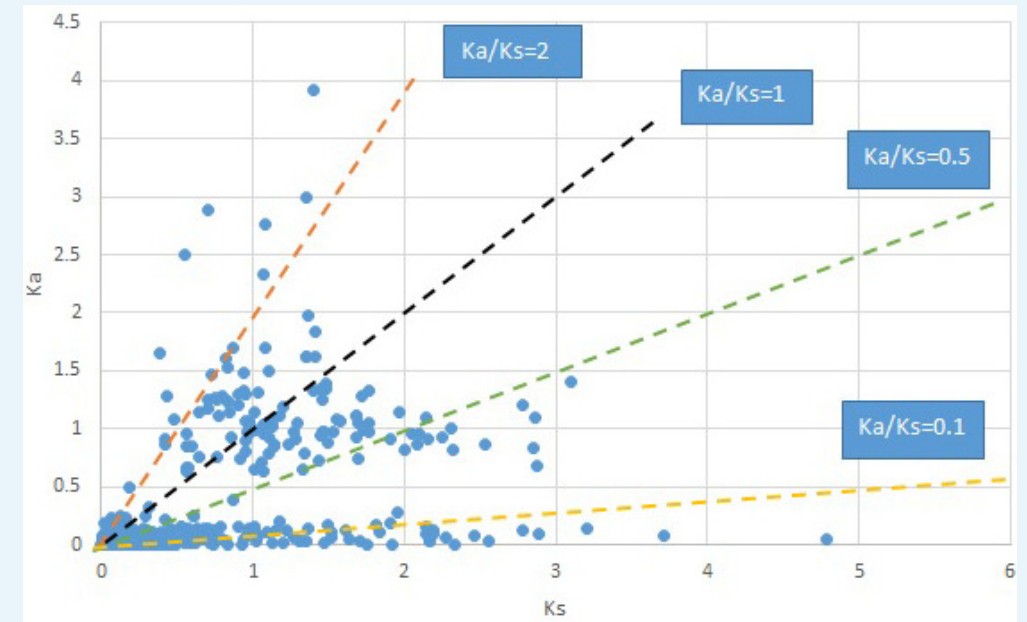

**Appendix 1—figure 3.** Plot of *Ka* and *Ks* values for each POG in 20 taxon pairs, in alphabetical order of genus names. *Acorus*.

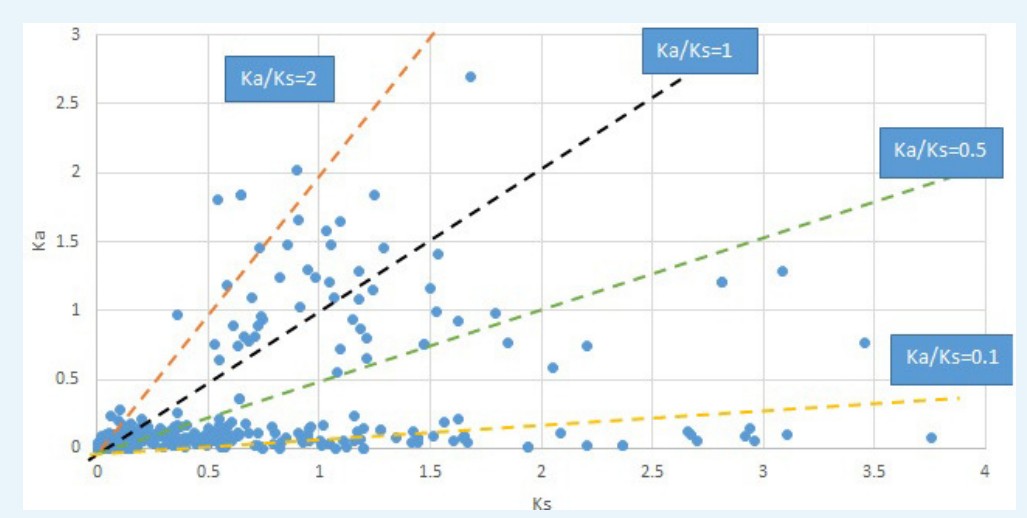

**Appendix 1—figure 4.** Plot of *Ka* and *Ks* values for each POG in 20 taxon pairs, in alphabetical order of genus names. *Calycanthus.*

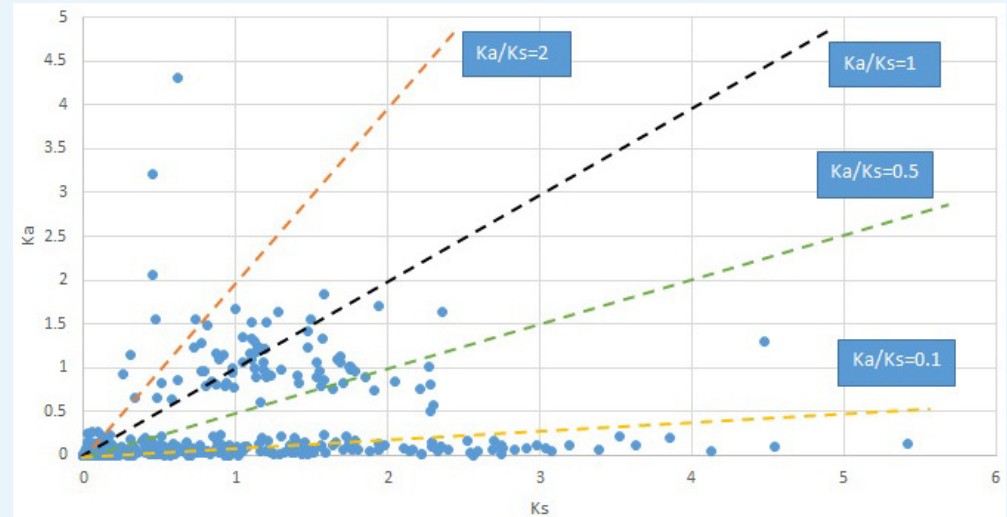

**Appendix 1—figure 5.** Plot of *Ka* and *Ks* values for each POG in 20 taxon pairs, in alphabetical order of genus names. *Campsis.*

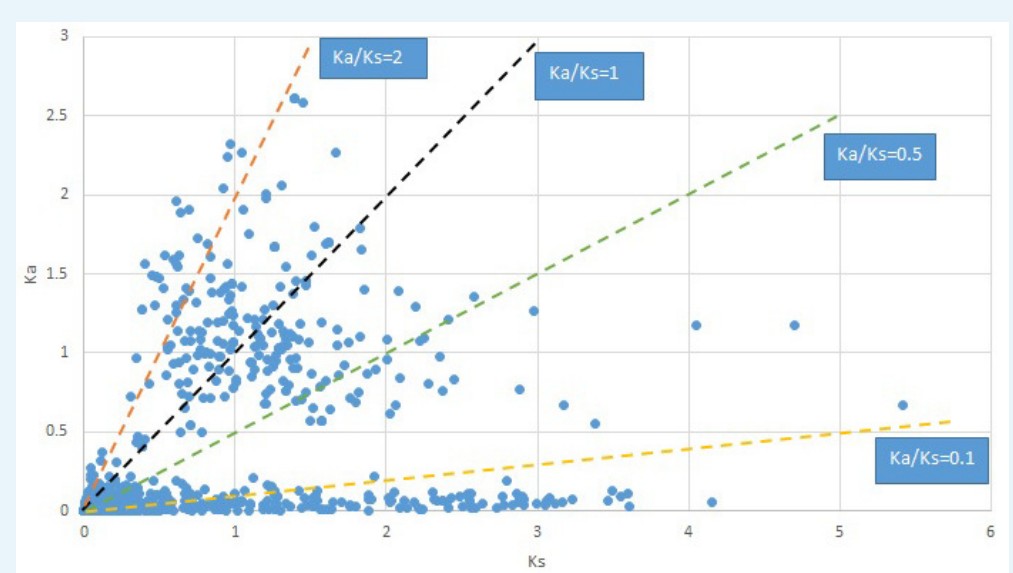

**Appendix 1—figure 6.** Plot of *Ka* and *Ks* values for each POG in 20 taxon pairs, in alphabetical order of genus names. *Convallaria*.

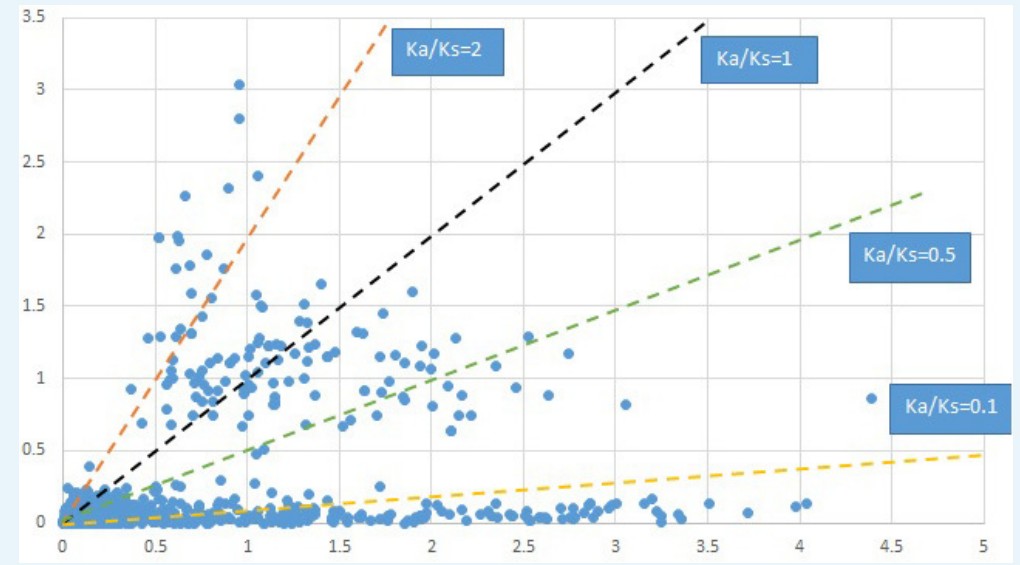

**Appendix 1—figure 7.** Plot of *Ka* and *Ks* values for each POG in 20 taxon pairs, in alphabetical order of genus names. *Cornus*-1.

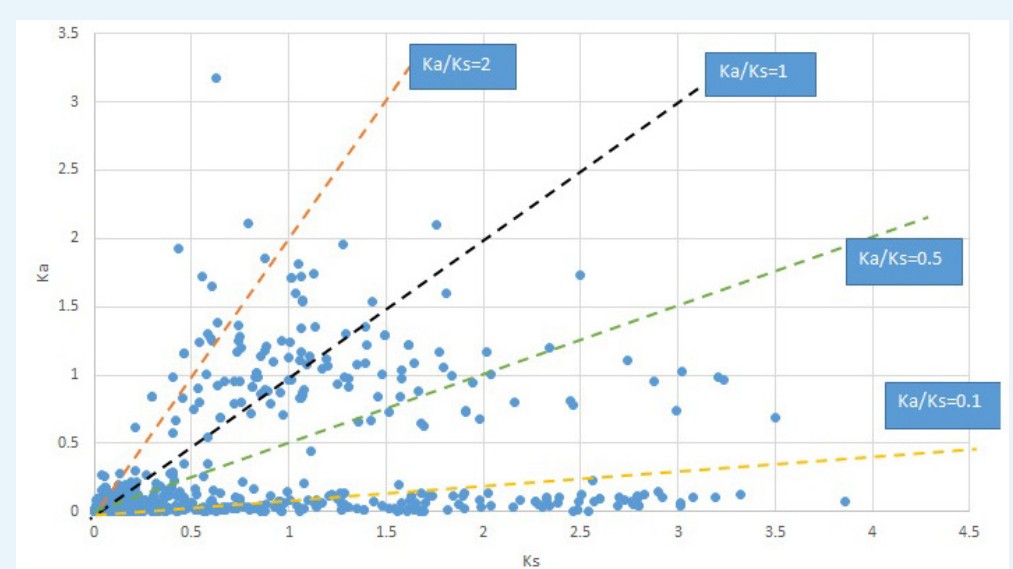

**Appendix 1—figure 8.** Plot of *Ka* and *Ks* values for each POG in 20 taxon pairs, in alphabetical order of genus names.  *Cornus*-2.

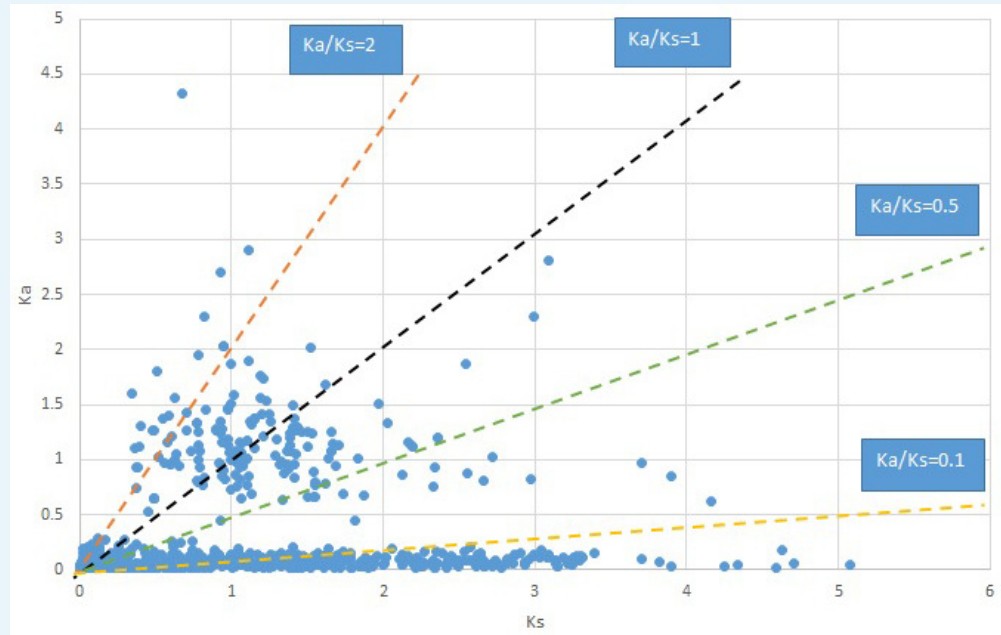

**Appendix 1—figure 9.** Plot of *Ka* and *Ks* values for each POG in 20 taxon pairs, in alphabetical order of genus names.  *Cotinus*.

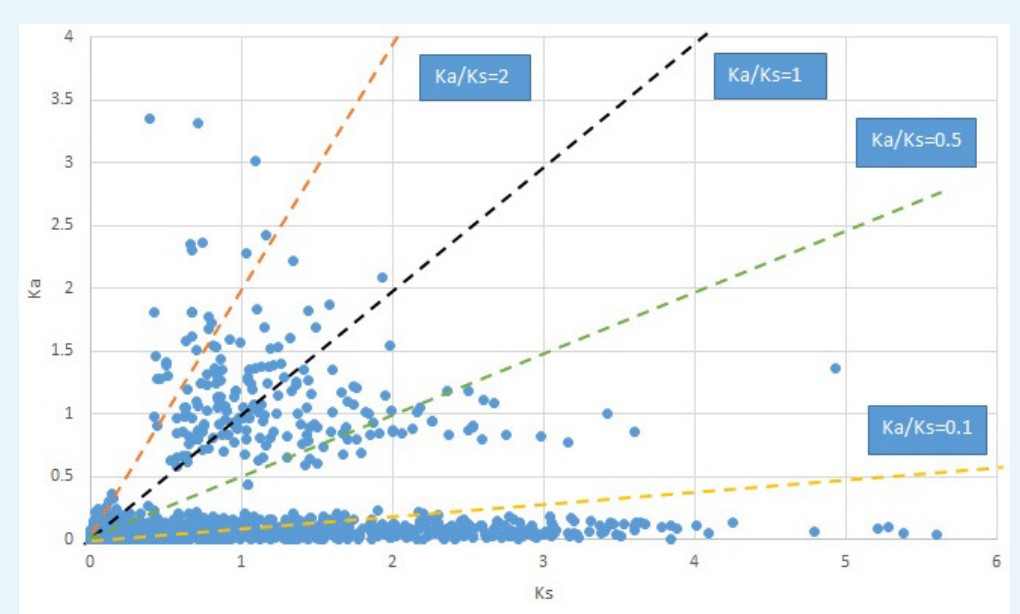

**Appendix 1—figure 10.** Plot of *Ka* and *Ks* values for each POG in 20 taxon pairs, in alphabetical order of genus names. *Croomia.*

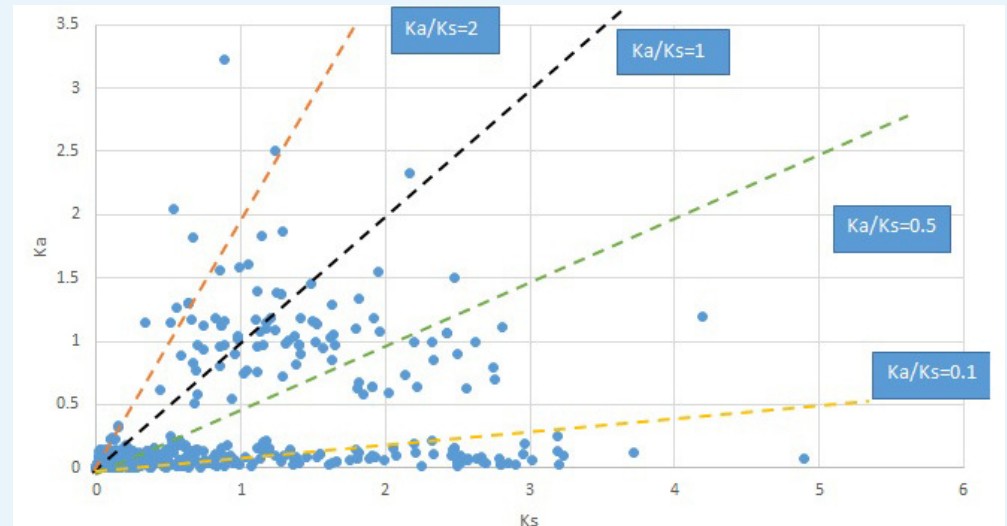

**Appendix 1—figure 11.** Plot of *Ka* and *Ks* values for each POG in 20 taxon pairs, in alphabetical order of genus names. *Dysosma.*

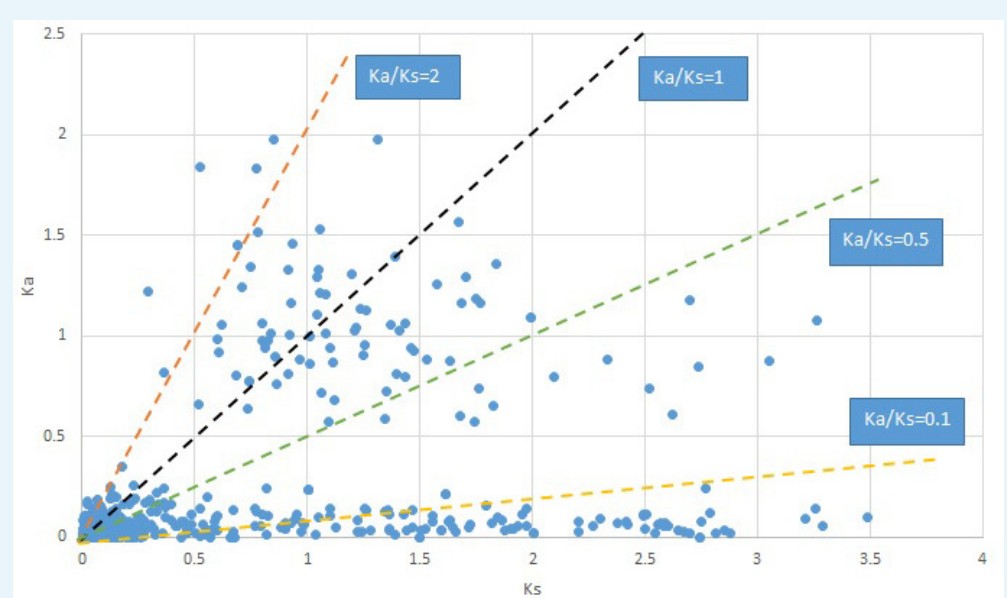

**Appendix 1—figure 12.** Plot of *Ka* and *Ks* values for each POG in 20 taxon pairs, in alphabetical order of genus names. *Gelsemium*.

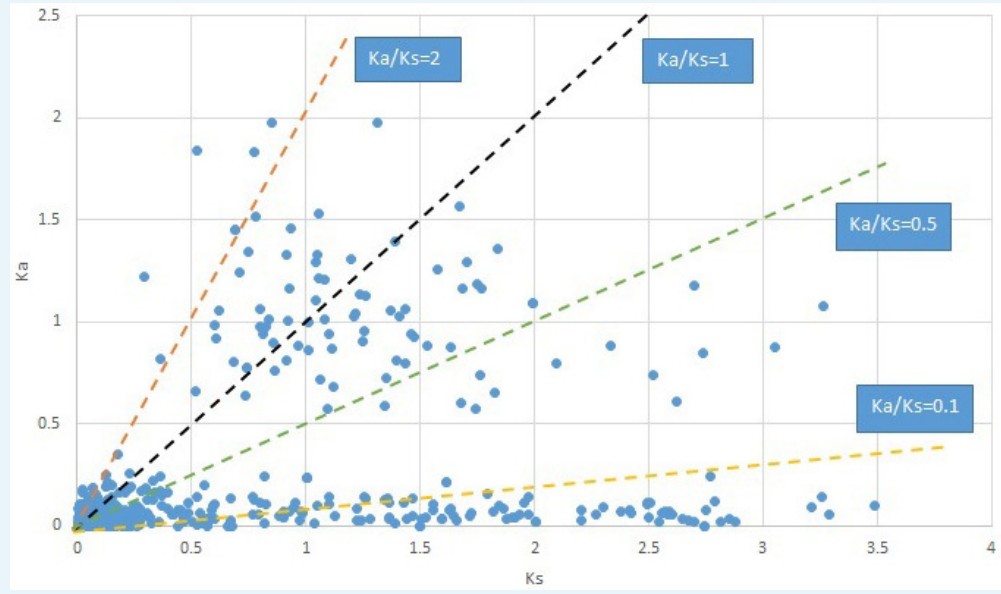

**Appendix 1—figure 13.** Plot of *Ka* and *Ks* values for each POG in 20 taxon pairs, in alphabetical order of genus names. *Hamamelis*.

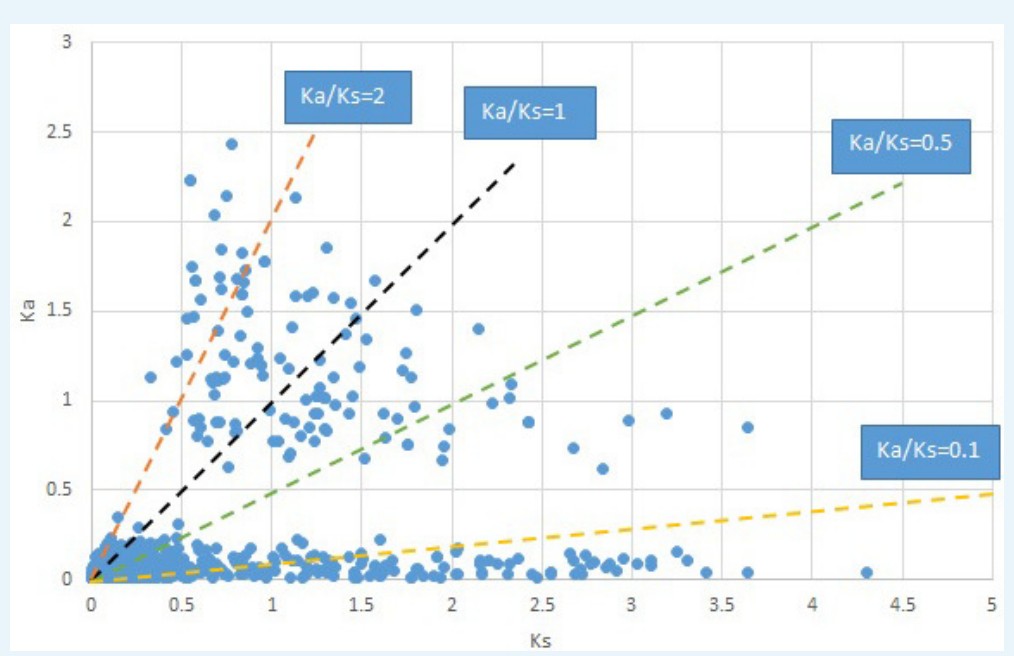

**Appendix 1—figure 14.** Plot of *Ka* and *Ks* values for each POG in 20 taxon pairs, in alphabetical order of genus names. *Liquidarnbar*.

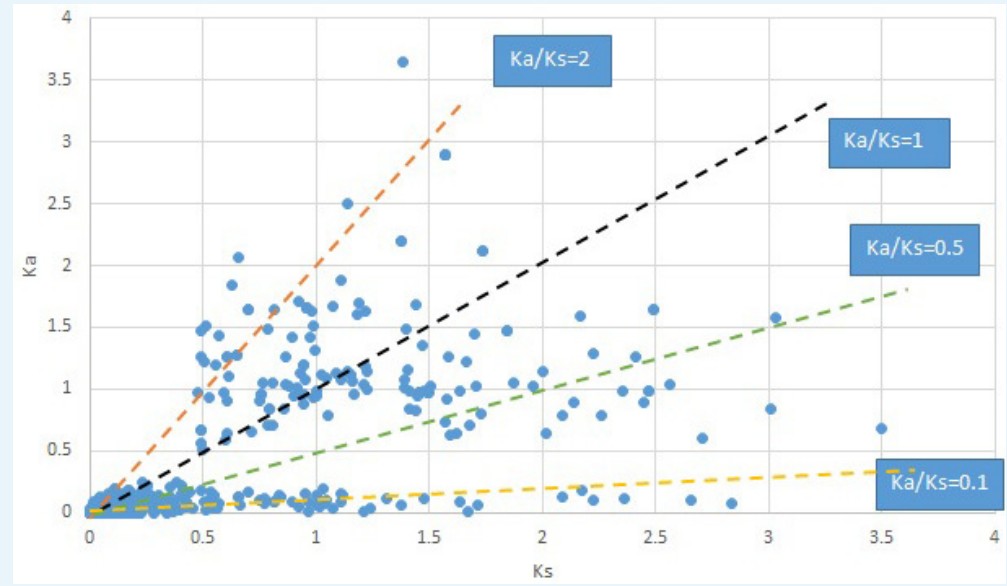

**Appendix 1—figure 15.** Plot of Ka and Ks values for each POG in 20 taxon pairs, in alphabetical order of genus names. *Liriodendron*.

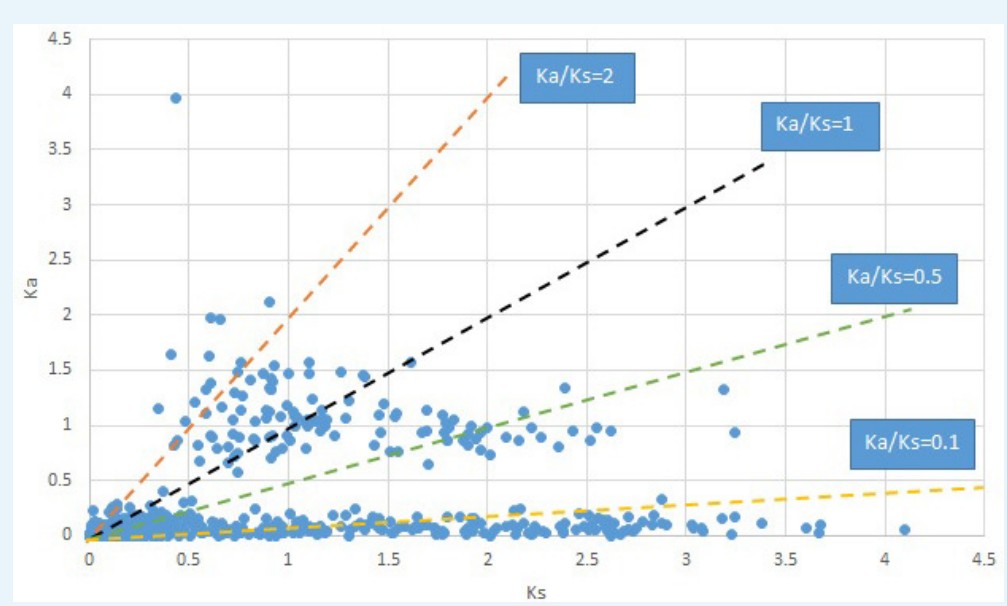

**Appendix 1—figure 16.** Plot of Ka and Ks values for each POG in 20 taxon pairs, in alphabetical order of genus names. *Meehania*.

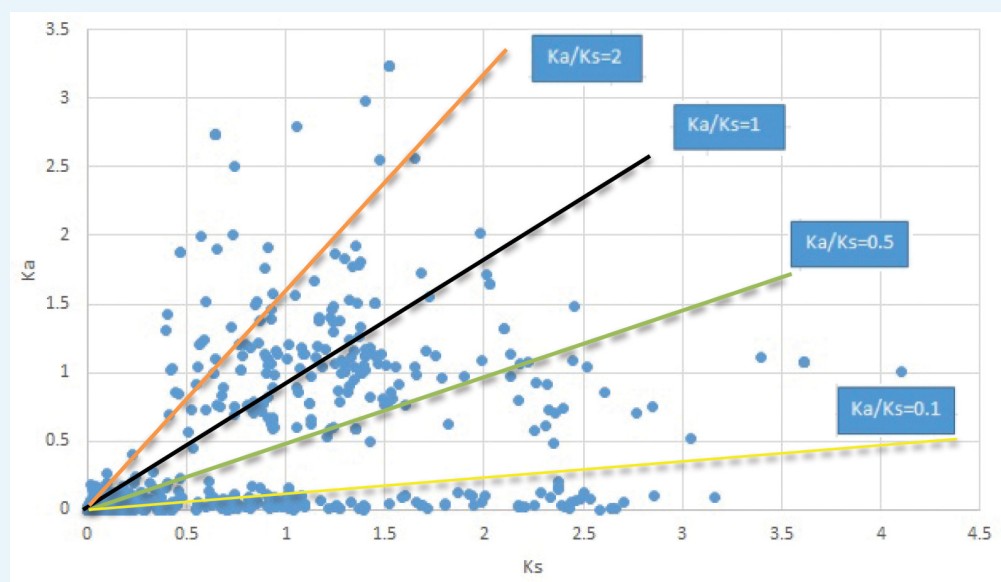

**Appendix 1—figure 17.** Plot of Ka and Ks values for each POG in 20 taxon pairs, in alphabetical order of genus names. *Menispermum*.

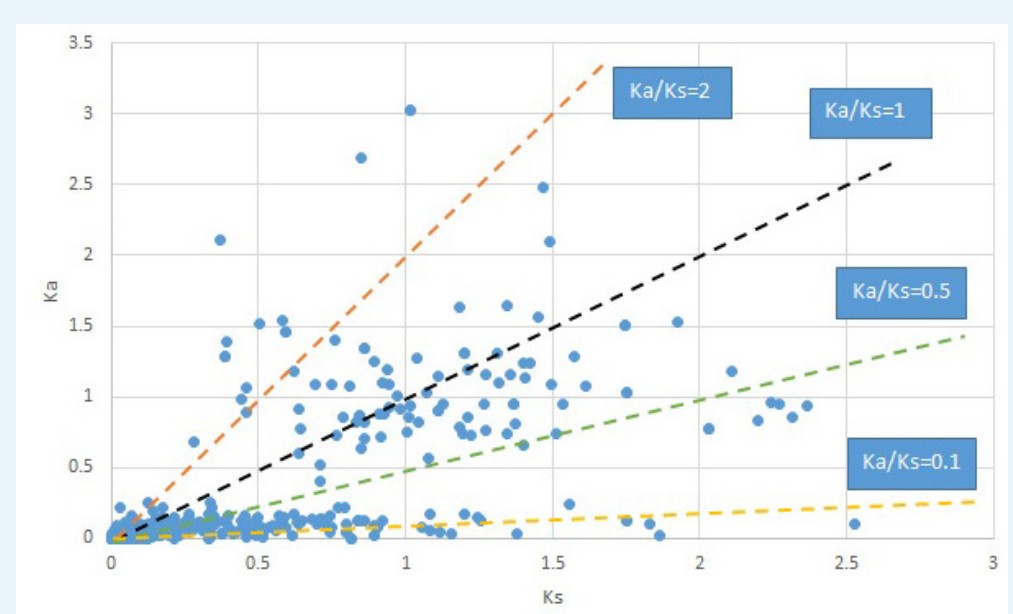

**Appendix 1—figure 18.** Plot of Ka and Ks values for each POG in 20 taxon pairs, in alphabetical order of genus names. *Nelumbo.*

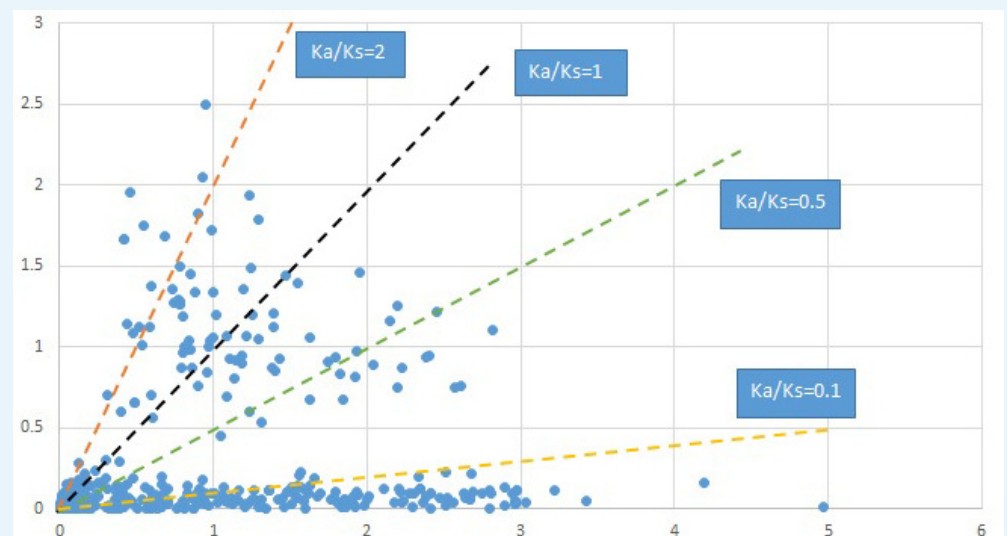

**Appendix 1—figure 19.** Plot of Ka and Ks values for each POG in 20 taxon pairs, in alphabetical order of genus names. *Penthorum.*

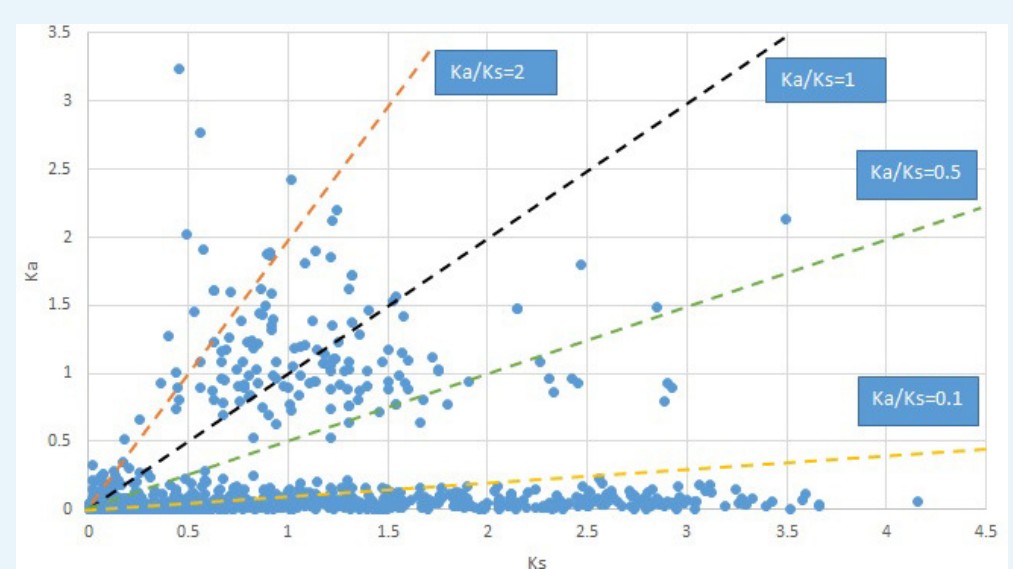

**Appendix 1—figure 20.** Plot of Ka and Ks values for each POG in 20 taxon pairs, in alphabetical order of genus names. *Phryma*.

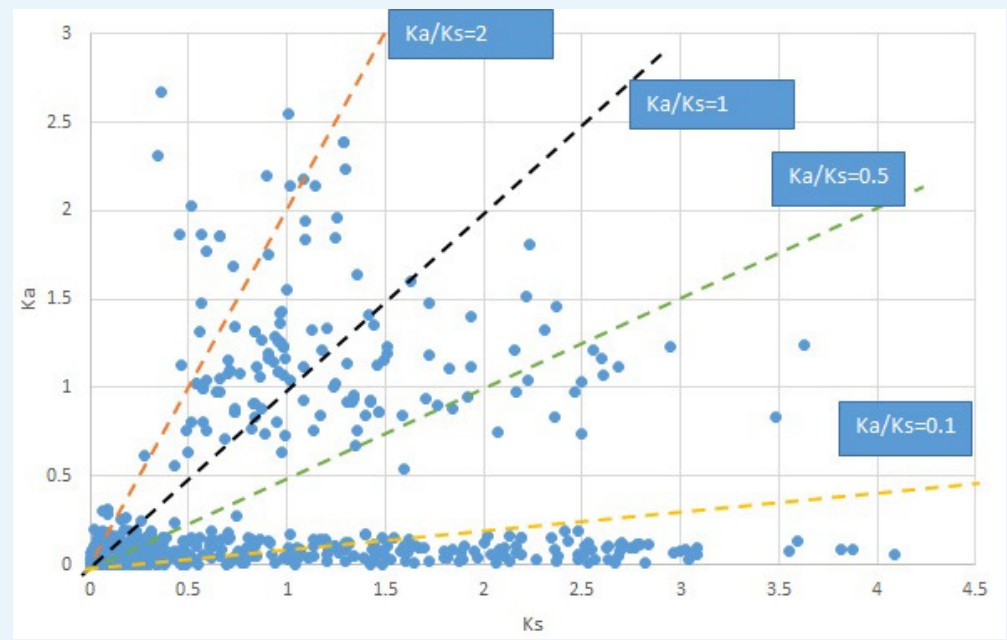

**Appendix 1—figure 21.** Plot of Ka and Ks values for each POG in 20 taxon pairs, in alphabetical order of genus names. *Sassafras*.

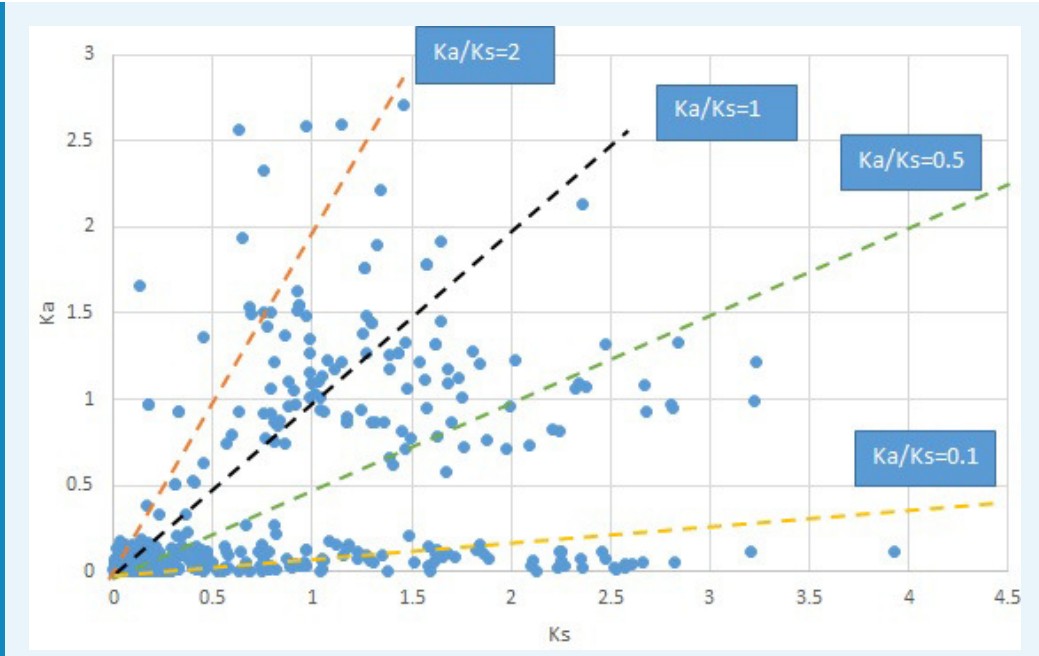

**Appendix 1—figure 22.** Plot of Ka and Ks values for each POG in 20 taxon pairs, in alphabetical order of genus names. *Saururus*.

