## [Decision Letter]

Thank you for submitting your article "Natural selection and repeated genome-wide patterns of molecular evolution following allopatric divergence" for consideration by *eLife*. Your article has been reviewed by three peer reviewers, one of whom is a member of our Board of Reviewing Editors, and the evaluation has been overseen by a Reviewing Editor and Patricia Wittkopp as the Senior Editor. The following individuals involved in review of your submission have agreed to reveal their identity: Vincent Savolainen (Reviewer #3).

The reviewers have discussed the reviews with one another and the Reviewing Editor has drafted this decision to help you prepare a revised submission.

Summary:

In this manuscript, the authors utilize comparative transcriptomics to estimate the role of evolution in singleton conserved genes following major geological events that lead to isolation by distance.

Essential revisions:

1) The utilization of *Ka/Ks* as a strict threshold to call positive selection across species with different evolutionary histories and separation times was considered potentially suspect and possibly creating some of the main conclusions. Some support for this claim and/or alternative approaches for these comparisons should be provided.

2) The observation by the authors that the relationship between *Ka* and *Ks* can invert at lower *Ks* is at some conflict with the published literature. The reviewers were wondering if this may be a bias in the data or if there are other avenues in the data that also support this. For example, does this trend hold up if individual POGs are followed across time?

3) A much tighter editing in the entire manuscript to reflect the potential bias that is incorporated via the use of POGs. The reviewers understood this choice to allow rate estimation, but it is still a biased set of genes that could be greatly influencing any and all evolutionary conclusions.

4) The figure presentation was not optimal and needs significant work to engage with the reader. I should note that all three reviewers commented on how the present figure presentations will undercut the potential impact of the manuscript.

Reviewer #1:

In this manuscript, the authors work to understand how geographic isolation has influenced genome wide evolution. To do this, they looked at the divergence of single-copy orthologues as these are the most directly comparable. This used transcriptome sequencing to identify the orthologues between species. I have some concerns about the methodology and how this may have influenced the claims and observations in this manuscript. These concerns are such that it is not clear how many of the claims and observations are linked to the use of POGs versus an approximation of the whole selective processes influencing these species.

I worry about the bias that using POGs introduces into the general conclusions derived from this work. POGs are by definition likely under a completely different selective schema than non-POGs. Previously work has shown that POGs are biased towards processes like developmental regulators and primary metabolism that are likely under purifying selection. In contrast, non-POGs are biased towards disease and biotic interaction loci that are potentially under different selective pressures. As such, it is not clear if using POGs provide any ability to discuss evolutionary processes at the species level. Instead, the discussion should be limited to what the POGs have experienced. At the start of the Discussion section, this limitation is implicit in that the discussion is always about the POGs. But then the discussion starts moving to whole genome and entire species level discussions without caveating that the POGs might provide a biased view. I don't think it is possible to use the POGs evolution to make any claims about whole genome events or whole species processes. The POGs are a list that is a priori biased towards genes that are under strong purifying selection and they exclude the majority of genes that may be under diversifying or other non-purifying selection processes. This caveat about what the use of POGs introduces as a selective bias should really be introduced throughout the Discussion section to better clarify what is or is not possible to state.

Additionally, this final sentence "Therefore, the divergence time values of the taxon pairs are still meaningful for the analysis of relationships between length of time of isolation and *Ks* value at peak frequency and between length of time of isolation and abundance of genes under different selection forces." illustrates that it is not clear if the authors consider the POGs as introducing a bias. The POGs are useful for estimating relative divergence time but absolute is fuzzier given the bias in selective events linked to POGs. Further, this bias in selective events means that it is only possible to compare selective forces on POGs and not selective forces on the whole genome.

The use of transcriptomics to sequence genes makes it hard to really ascertain if the identified positive selection is really positive or possibly an artifact of duplication, sub-functionalization and inaccurate POG identification. Some ascertainment of bias the transcriptome introduces above genome sequence identification of POGs would be needed to determine how much technical vs biological signal is present in the positive selection data. There is some effort to address these concerns in the Discussion section however this section is separate from the rest of the discussion so that a reader would have to make it to the end to realize that there are concerns. Are there no existing transcriptome datasets with matched genomes that the authors could use to estimate technical bias in calling POGs?

There are some weird call outs on Materials and methods that make some aspects hard or impossible to review. For example, in the Results section, the authors state "(for method of the mapping analysis, see caption of Table 1)" yet the caption of table 1 has no real information on the mapping analysis approaches.

The figures in excel are a bit hard to read. This is largely because the use of dots on the lines really eliminates the ability to compare traces (See Figure 3 for an example). The authors should consider using simply color for the lines and possibly moving to a higher resolution figure generator.

Reviewer #2:

The manuscript describes a study examining how 1-to-1 orthologous genes in 20 allopatric species pair diverge after species divergence. Specifically, the authors indicate that it remains an open question "how genome [have diverged] over time after geographic isolation has halted gene flow". The major findings include: (1) >90% of genes are under purifying selection, (2) divergence time estimates, which are based on synonymous substitution rates, correlates positively and negatively with proportions of genes under moderate and strong purification selection, respectively, (3) 200 genes under strong positive selection, few are shared across species.

The authors reasoned that findings (1, 3) may indicate maintenance of a balance between the ability to conserve ancestral functions and the ability to evolve new features beneficial for new adaptations. In some of the earliest cross-species comparative genomics studies in early 2000s, this has been noted. Thus, it is not clear what the significance of this finding is beyond the fact that the specific species have not been examined in this context. The connection between geographic isolation over time and the evolutionary rate of genes – initially genes are experiencing strong purifying selection, then this selection is relaxed – is intriguing. But there is one line of evidence supporting this claim, indirectly. I have suggested alternative approach that may be more informative. Below are details on the above points as well as other thoughts on the manuscript.

- Results section: It is too simplistic to use the *Ka/Ks* value over a threshold to call genes that are under positive selection. Particularly, as the authors pointed out, the species pairs have diverged quite recently where "most genes" (define what most means) have *Ka* < 0.04. The fewer the sites are used to infer substitution rates, the corresponding variances for the estimates will be larger and render the estimates unreliable. Thus, the criteria for calling genes with positive selection needs to be more stringent and the authors need to incorporate considerations of variance of estimates to provide statistical support for their calls. This is a point the authors concur in their discussion. But by just removing extreme values, the underlying issue on the reliability of estimates remain unaddressed.

- Related to the point above, in the Results section, the authors conducted gene set enrichment analysis of potentially positively selected genes. Given I am not convinced that the genes the authors focused on are truly under position selection, I cannot evaluate the results of enrichment analysis.

- Results section, Figure 5: The authors use BEAST to infer divergence time and then try to determine the relationships between inferred divergence time and *Ks* – it is not clear what the purpose is as these two estimates are confounding. My understanding is that BEAST uses prior probability based on sequence evolution rate estimates to infer posterior probability of a divergence time estimate. The underlying assumption and considerations for *Ks* estimation and the evolutionary rate considered as prior in BEAST are similar. So, these two sets of values are dependent. In the, the authors went on and suggest that this pattern "revealed a repeated genome-wide pattern of divergence of POGs among the taxon pairs in allopatric speciation…" Given the confounding nature of *Ks* and divergence time estimate, I am not sure if such conclusion is justified.

- Results section, Figure 6: here is the core finding for the authors' claim on the relationships between divergence time and selection pressure. It is clear that, using these species pairs, the earlier diverging (peak *Ks*, not sure why the more accurate divergence time estimate not used here) ones tend to have larger proportion of genes under stronger selection. The effect size of the correlation is rather remarkable. One unclear aspect of the analysis is whether the genes examined are the same across species pairs. It was noted that there are few orthologous genes that are 1-to-1 among most species pairs. Thus, it is likely the analysis is done using all available pairs. If that is the case, how should one compare one species pair to another?

- Related to the point above, the patterns based on proportions are difficult to interpret. A direct way to demonstrate the authors' point is by examining the evolution rate of EACH orthologous gene pair and show that a gene experienced stronger purifying selection initially than later on the selection was relaxed. Since this is central to the authors' major point, this should be demonstrated. In addition to potentially providing support for the authors' claim, it will provide some resolution to the study in which genes contribute to the patterns, particularly if the authors follow up with gene set enrichment analysis on those exhibit the expected patterns.

- Discussion section: The authors discussed a few of the caveats in the study which is helpful. Aside from the point on rate estimate raised earlier, I am not sure that I agree with the authors' assertion that "the likelihood of non-orthology of these genes is probably very low…" using transcriptome-based inference. The arguments are not supported by quantitative information and is hand-wavy. It will be useful to bring in comparative genomic studies as reference points. Otherwise, it remains a potential thorny issue in their methodology.

Reviewer #3:

The paper looks at the evolution of genes in pairs of species that have been isolated geographically. To do so, the authors selected 20 pairs that have a South East Asia versus North America disjunction and calculated the ratio of nonsynonymous versus synonymous substitutions across their transcriptomes. This is a clever approach. Strikingly, they found some sort of constancy in the results, with that >90% of the genes examined being under purifying selection and <10% being under positive selection. This is what we would expect, but it is a nice way to show it, using comparative analyses and large datasets. I would think people have looked at this in model organisms (mice, humans), but not throughout wild plant species. I think it would definitely make a great contribution to *eLife*, although I think the author could make more of a general point by adding other pairs of taxa, such as those from a North to South America disjunction, or South America versus Africa disjunction, or maybe even using species that have colonised island versus mainland sisters. I think the text could be made clearer to the readers, e.g. explaining, what is the expectation of *Ka/Ks* ratio under positive or purifying selection (I would also use *Kn* rather than *Ka* for nonsynonymous substitutions). I noted there are some custom scripts used here, have they been made available as supplementary information or in a public repository? For the dating exercise, did I understand correctly that the tree was built and dated with BEAST using 7 POGs shared by all taxa, and then the dates were plotted against *Ks* for the other genes to avoid circularity? Finally, I think the figures could be made much nicer (artwork); there are also many inconsistencies in the reference section; but I would recommend acceptance pending concerns above have been taken into account.

---

## [Author Response]

Summary:In this manuscript, the authors utilize comparative transcriptomics to estimate the role of evolution in singleton conserved genes following major geological events that lead to isolation by distance.Essential revisions:1) The utilization of Ka/Ks as a strict threshold to call positive selection across species with different evolutionary histories and separation times was considered potentially suspect and possibly creating some of the main conclusions. Some support for this claim and/or alternative approaches for these comparisons should be provided.

Thanks for these suggestions. We will further clarify the effectiveness of *Ka/Ks* in examining the pattern of variation of selection pressure on genes and comparing it among lineages in the revised manuscript. Previous studies have indicated that data from the various lineages provide parallel evidence on molecular evolution following allopatric divergence. A potential concern, though, may be the accuracy of *Ka/Ks* estimation (e.g., potential saturation of *Ks* in taxon pair separated by older times and very small *Ks* value in taxon pairs separated recently – leading to inflated *Ka/Ks*). We cleared this concern in the manuscript (please see methods and discussion in line 637-666). For the time scale of divergence across species pairs, we have explained that saturation of *Ks* is unlikely or rare (please see discussion section). We removed *Ks* = 0 data points in all of the analyses. Furthermore, we examined the distribution of *Ks* for genes with *Ka/Ks* >2 and found the *Ks* values varied in a range, not clustered near zero).

We provide support for the claim of *Ka/Ks* to identify positive selection and alternative approaches for comparisons as below:

1) The main finding is the pattern observed among lineages and identification of a small subset of genes under strong positive selection with *Ka/Ks* >2. Comparison of the rates of fixation of synonymous and nonsynonymous mutations provides a powerful tool for understanding the mechanisms of DNA sequence evolution. The ratio of nonsynonymous (*Ka* or *dN*) to synonymous (*Ks* or *dS*) nucleotide substitutions is considered probably the most common method for identifying positive selection (*Ka/Ks*>1) (Yang, 2003) in molecular evolutionary studies. The method is known to be a conservative approach to call positive selection on a gene as it uses an average *Ka/Ks* ratio across all nucleotide sites. It is possible that some genes have an overall *Ka/Ks* <1 but some sites might have positive selections with *Ka/Ks* >1. This limitation does not affect the comparison of the pattern of variation of molecular divergence across lineages. The ratio of *Ka/Ks* values were used to partition the genes into arbitrary categories different in selection pressures.

2) For identification genes under positive selection, we took caution to call only those with *Ka/Ks*>2 and annotated them to provide candidates under strong positive selection for future studies. We did revise to apply an alternative method (variation cluster) to confirm signals of positive selections in the candidate genes (Wagner, 2007; see more below).

3) The Yn2000 model in codeml of PAML applied in the study is a maximum likelihood method for estimation of *dN* and *dS*. As stated by the authors “the ML method, which accounts for both the transition bias and the codon usage bias, should be the preferred method for estimating *dS* and *dN* between two sequences.” (Yang and Nielson, 2000). This method has been widely used for detecting positive selection in genes. Another common method for detecting positive selection is examining the allele frequency spectrum of the gene in the population. – this is based on population data, not feasible to our study.

4) “Two broad classes of approaches exist to identify positive selection (Kreitman, 2000; Bamshad and Wooding, 2003). They both rely on predictions made by the neutral theory of molecular evolution (Kimura, 1983). The first approach compares the incidence of two different classes of genetic change within genes (Li, 1997; Kreitman, 2000), synonymous (silent) changes, which are likely to be neutral, and nonsynonymous or amino acid replacement changes, which are more likely subject to selection. Specifically, the ratio *N/S* of the number of nonsynonymous (*N*) to synonymous (*S*) changes per gene, or the ratio *K*_a_/*K*_s_ of the fraction of nonsynonymous (*K*_a_) to synonymous changes (*K*_s_) per nonsynonymous and synonymous site, can give an indication of positive selection. A ratio *K*_a_/*K*_s_ significantly greater than one, for example, indicates an excess of amino acid replacement substitutions over (neutral or weakly selected) silent substitutions. It indicates positive selection. Many variations of this class of test exist. They differ in the amount of sequence data and computational resources required (Suzuki and Gojobori, 1999; Suzuki, 2004; Massingham and Goldman, 2005; Pond and Frost, 2005; Zhang et al., 2005). The second class of tests relies on predictions made by the neutral theory for allele or haplotype frequencies (Kreitman, 2000; Bamshad and Wooding, 2003) within and among populations. For example, in a genomic region where positive selection has swept a mutation to high frequency, one would expect a low amount of sequence diversity, an excess of rare alleles, and a greater amount of linkage disequilibrium than predicted by the neutral theory (Bamshad and Wooding, 2003). Selection acting on one population but not others can lead to a greater than expected degree of population differentiation. Test statistics, such as Tajima's *D*, Fu's *W*, Wright's *F*_ST_, and many others, exploit information in these patterns (Fu, 1996; Tajima, 1989; Kreitman, 2000). The distinction between such tests is not sharp, and some tests (McDonald and Kreitman, 1991) arguably fall into both categories.” (Wagner, 2007).

5) "Rapid Detection of Positive Selection in Genes and Genomes Through Variation clusters?” was developed for detecting positive selection for pairs of sequences by Wagner, (2007). It “…detects *variation clusters* of aggregated nucleotide substitutions that are too closely spaced to be observed by chance alone and that thus violate the predicted distribution of substitution spacing for neutral variation.”. We ran this method for genes with *dN/dS* > 2 for all species pairs to see if how many of them contain significant signals of substitution clustering.

2) The observation by the authors that the relationship between Ka and Ks can invert at lower Ks is at some conflict with the published literature. The reviewers were wondering if this may be a bias in the data or if there are other avenues in the data that also support this. For example, does this trend hold up if individual POGs are followed across time?

We observed a weak negative trend between *Ks* and the enrichment of *Ka/Ks* genes with *Ka/Ks*>1, suggesting as divergence time increased, enrichment of this category of genes may decrease due to the change of selection pressure on some genes. This may be a result of accumulation of synonymous substitution in the gene.

We did not observe revert relationship between *Ka* and *Ks*. We observed that at low *Ks* (recent divergence time within 10 million years) and at the range of *Ka/Ks* within 0.5 or below, the relationship between the gene enrichment and time is reversed (positive for 0.1 <*Ka/Ks* <0.5; negative for *Ka/Ks*<0.1). So we speculated an explanation for these observations: some genes relaxed the purifying selection force from strong force. We do not have direct evidence to support this within a species pair. We examined the POG shared by different taxa diverged at different times as suggested by the reviewer 3, and did find that some genes with *Ka/Ks* <0.5 (or regardless the *Ka/Ks* values) showed a positive relationship between the *Ka/Ks* values and the divergence time. But many did not show a correlation. The common genome-wide pattern of POG evolution is likely an outcome of both gene and taxon specific effects.

3) A much tighter editing in the entire manuscript to reflect the potential bias that is incorporated via the use of POGs. The reviewers understood this choice to allow rate estimation, but it is still a biased set of genes that could be greatly influencing any and all evolutionary conclusions.

We revised the text extensively and made clear that the study examined one set of genes and the pattern found was a “genome-wide” pattern of these genes. We reorganized the manuscript and moved the discussion of caveats of the study to the front. An inspection of our transcriptome data showed that a majority (~60%-80%) of the ortholog groups (gene families) identified in our study were the POGs examined, indicating the POGs represent a significant portion of the transcriptome. We added this information to the manuscript.

4) The figure presentation was not optimal and needs significant work to engage with the reader. I should note that all three reviewers commented on how the present figure presentations will undercut the potential impact of the manuscript.

Thanks for the comments. We have redrawn the first three figures to break down the composite figure of all lineages to 20 individual small figures. We think the new figures showed much clearer distribution of the *Ka, Ks*, and *Ka/Ks* values in each taxon pair.

Reviewer #1:In this manuscript, the authors work to understand how geographic isolation has influenced genome wide evolution. To do this, they looked at the divergence of single-copy orthologues as these are the most directly comparable. This used transcriptome sequencing to identify the orthologues between species. I have some concerns about the methodology and how this may have influenced the claims and observations in this manuscript. These concerns are such that it is not clear how many of the claims and observations are linked to the use of POGs versus an approximation of the whole selective processes influencing these species.I worry about the bias that using POGs introduces into the general conclusions derived from this work. POGs are by definition likely under a completely different selective schema than non-POGs. Previously work has shown that POGs are biased towards processes like developmental regulators and primary metabolism that are likely under purifying selection. In contrast, non-POGs are biased towards disease and biotic interaction loci that are potentially under different selective pressures. As such, it is not clear if using POGs provide any ability to discuss evolutionary processes at the species level. Instead, the discussion should be limited to what the POGs have experienced. At the start of the Discussion section, this limitation is implicit in that the discussion is always about the POGs. But then the discussion starts moving to whole genome and entire species level discussions without caveating that the POGs might provide a biased view.

Thanks for the reviewer referring to previous work reporting annotated gene functions of probably true single copy genes based on genome information. Some or many of the genes we studied may represent non-single copy genes (or low copy gene families). Our annotation of genes under positive selection revealed some of the genes were involved in biological processes responding to environmental stimuli.

Previous studies showed that the genome does not evolve in a homogeneous manner. Different type of genes and different portion of the genome may experience different evolutionary processes. Purifying selection, positive selection, and neutral evolution are all evolutionary processes. Evolutionary process is not restricted to positive selection. Our study examined molecular divergence and evolution of POGs to provide a view of divergence of the species in these genes. Thanks, the reviewer reminding that we take care to be clearer about the interpretation. Evolution of non-POGs can be examined in the future through analyses of each gene families across the genome to understand the evolution of those genes. Together, the data will provide a more complete view of molecular evolution of these species. We revised text to make clear about this point in the manuscript.

I don't think it is possible to use the POGs evolution to make any claims about whole genome events or whole species processes. The POGs are a list that is a priori biased towards genes that are under strong purifying selection and they exclude the majority of genes that may be under diversifying or other non-purifying selection processes. This caveat about what the use of POGs introduces as a selective bias should really be introduced throughout the Discussion section to better clarify what is or is not possible to state.

As mentioned above, the POGs represent a majority of the orthologous gene families in the transcriptome. We made clear that the inferred processes on the POGs represent a small window of the genome process.

Additionally, this final sentence "Therefore, the divergence time values of the taxon pairs are still meaningful for the analysis of relationships between length of time of isolation and Ks value at peak frequency and between length of time of isolation and abundance of genes under different selection forces." illustrates that it is not clear if the authors consider the POGs as introducing a bias.

We revised the text to more clearly express what we attempted to say.

The POGs are useful for estimating relative divergence time but absolute is fuzzier given the bias in selective events linked to POGs.

The *Ks* at peak value is predictive of the relative timing of taxon pair divergence. The absolute time of divergence was estimated using a subset of POGs shared among taxa on a global phylogeny of all taxa. In literature, most phylogenetic studies and dating analyses use genes with functions that are likely under various degrees of purifying selection. The potential bias in divergence time dating analysis in our study is expected to be across the board and does not affect the relative timing among taxa. We revised text to be clear about this.

We thought over the reviewer’s comment here. It might be that if selection makes the genes inappropriate for dating divergence due to selection, neither relative time nor absolute time can be estimated since selection can be dramatically different among the genes.

Further, this bias in selective events means that it is only possible to compare selective forces on POGs and not selective forces on the whole genome.

Thanks. It is correct. We now made this clearer in writing and clarify it in the Introduction.

The use of transcriptomics to sequence genes makes it hard to really ascertain if the identified positive selection is really positive or possibly an artifact of duplication, sub-functionalization and inaccurate POG identification.

For recently diverged sister species, we feel that the probability of orthologous copy being expressed in the same organ in both species is high. Most of the genes with *Ka/Ks* > 2 were also found to be under positive selection by the variation cluster analaysis (see above). The differences between the two methods may be a result of differences in the principle.

Some ascertainment of bias the transcriptome introduces above genome sequence identification of POGs would be needed to determine how much technical vs biological signal is present in the positive selection data. There is some effort to address these concerns in the Discussion section however this section is separate from the rest of the discussion so that a reader would have to make it to the end to realize that there are concerns. Are there no existing transcriptome datasets with matched genomes that the authors could use to estimate technical bias in calling POGs?

We have reorganized the information and moved the discussion on bias up front in the introduction. Bias introduced by transcriptome sequences is likely low given the relative recent time of divergence of sister species. The transcriptomes were deeply sequenced to recover both low and highly expressed transcripts. We made clear we were comparing putative SCs. Unfortunately, we do not have genome sequences of any taxon pairs to evaluate how many of the POGs are truly single copy.

There are some weird call outs on Materials and methods that make some aspects hard or impossible to review. For example, in the Results section, the authors state "(for method of the mapping analysis, see caption of Table 1)" yet the caption of table 1 has no real information on the mapping analysis approaches.

Sorry about this error. I have now added the methods to the Materials and methods section. It is also described below: To see how many of these POGs were represented in the “global” single or low copy gene data sets of seed plants (Li et al., 2017; identified from a combination of genome and transcriptome data) and flowering plants (De Smet, 2013; identified from 20 angiosperm genomes), respectively, we performed the following mapping analyses. First, the single-copy or low-copy gene names *of Arabidopsis thaliana* shared by seed plants or flowering plants were downloaded from the respective publications. Second, the dataset including all cDNA sequences of *Arabidopsis thaliana* was downloaded from its genome database (TAIR10: https://www.arabidopsis.org/). Then the cDNA sequences of the single-copy or low-copy gene of *Arabidopsis thaliana* shared by seed plants or flowering plants were selected out from the dataset of all cDNA using gene names by Perl script. Finally, the cDNA sequences from each of the 20 taxon-pairs were mapped against the selected gene cDNA sequences of *Arabidopsis thaliana* by Blast program, and those that were mapped by the cDNA from each of the 20 taxon pairs were selected and recorded by Perl script.

The figures in excel are a bit hard to read. This is largely because the use of dots on the lines really eliminates the ability to compare traces (See Figure 3 for an example). The authors should consider using simply color for the lines and possibly moving to a higher resolution figure generator.

We have made new Figure 1, Figure 2 and Figure 3 to make information clear. We think Figure 5 is fine to the purpose. So, we did not change Figure 5.

Reviewer #2:The manuscript describes a study examining how 1-to-1 orthologous genes in 20 allopatric species pair diverge after species divergence. Specifically, the authors indicate that it remains an open question "how genome [have diverged] over time after geographic isolation has halted gene flow". The major findings include: (1) >90% of genes are under purifying selection, (2) divergence time estimates, which are based on synonymous substitution rates, correlates positively and negatively with proportions of genes under moderate and strong purification selection, respectively, (3) 200 genes under strong positive selection, few are shared across species.The authors reasoned that findings (1, 3) may indicate maintenance of a balance between the ability to conserve ancestral functions and the ability to evolve new features beneficial for new adaptations. In some of the earliest cross-species comparative genomics studies in early 2000s, this has been noted. Thus, it is not clear what the significance of this finding is beyond the fact that the specific species have not been examined in this context.

We are not sure which papers the reviewer was referred to. We indicated in the manuscript that although the pattern might have been noted earlier in other organisms, here we provide a set of empirical data from multiple lineages of flowering plants supporting the idea. Significance of the study is now more clearly stated in our Abstract and Conclusion.

The connection between geographic isolation over time and the evolutionary rate of genes – initially genes are experiencing strong purifying selection, then this selection is relaxed – is intriguing. But there is one line of evidence supporting this claim, indirectly. I have suggested alternative approach that may be more informative. Below are details on the above points as well as other thoughts on the manuscript.

Thanks for the suggestions. We have revised accordingly. Please see below.

- Results section: It is too simplistic to use the Ka/Ks value over a threshold to call genes that are under positive selection. Particularly, as the authors pointed out, the species pairs have diverged quite recently where "most genes" (define what most means) have Ka < 0.04. The fewer the sites are used to infer substitution rates, the corresponding variances for the estimates will be larger and render the estimates unreliable. Thus, the criteria for calling genes with positive selection needs to be more stringent and the authors need to incorporate considerations of variance of estimates to provide statistical support for their calls. This is a point the authors concur in their discussion. But by just removing extreme values, the underlying issue on the reliability of estimates remain unaddressed.- Related to the point above, in in the Results section, the authors conducted gene set enrichment analysis of potentially positively selected genes. Given I am not convinced that the genes the authors focused on are truly under position selection, I cannot evaluate the results of enrichment analysis.

We call the genes with *Ka/Ks* >? as putative positive selection genes in the discussion. We also make a note that some of these may be false due to small *Ks* values. Average *Ka/Ks* value over the entire gene sequences is considered an extreme conservative method in detecting gene positive selection. Species recently diverged do not have the problem of *dS* saturation. The concern is the nearly zero value of *dS* or there are very few sites with nucleotide substitutions. Therefore, removing extreme small *dS* values is appropriate to avoid inflation of *dN/dS* values from small *dS*. We examined the *dS* values of POGs with *Ka/Ks* >2 in each taxon pair and they varied in a wide range (only a tiny portion are <0.05; see Supplementary file 5, Supplementary file 6; newly added). Our data also show that all of the taxon pairs have *Ks* in peak frequency >=0.003, except one pair of varieties (Supplementary file 3) and a small portion of POGs with *Ka/Ks* >2 have *Ks* <0.005). We also manually checked the sequence alignment of genes with *Ka/Ks* >2 for two taxon pairs (*Cornus*-2 and *Acorus*) and in general observed multiple substitutions in each pair of sequences (results not shown; available upon request). We further used variation cluster to check on these positive selection genes (see above). Although variance of *dN/dS* ratios may be large in a gene with few substitutions, an average of *Ka/Ks* > 1 indicates excess nonsynonymous substitutions than expected from neutral evolution if *Ks* is not too small.

- Results section, Figure 5: The authors use BEAST to infer divergence time and then try to determine the relationships between inferred divergence time and Ks – it is not clear what the purpose is as these two estimates are confounding. My understanding is that BEAST uses prior probability based on sequence evolution rate estimates to infer posterior probability of a divergence time estimate. The underlying assumption and considerations for Ks estimation and the evolutionary rate considered as prior in BEAST are similar. So, these two sets of values are dependent. In the, the authors went on and suggest that this pattern "revealed a repeated genome-wide pattern of divergence of POGs among the taxon pairs in allopatric speciation…" Given the confounding nature of Ks and divergence time estimate, I am not sure if such conclusion is justified.

The reviewer may misunderstand the analyses and the points. The “repeated genome-wide pattern of divergence of POGs among the taxon pairs in allopatric speciation…" refers to the relative enrichment/abundance of genes under different selection pressures, not relative to the divergence time estimations.

BEAST uses both *dN* and *dS* in estimating the divergence time of a portion of the genes shared by all the taxa (7 and 79 respectively). The purpose of comparing the divergence time and *dS* in peak frequency (in considering all POGs in a species pair, not just those shared among all taxon pairs) was to see if *dS* values that are most abundant in a species pair reflect the relative level of species isolation. The two data sets are not confounding. The genes with *dS* in peak frequencies are not the same genes used for divergence time dating.

- Results section, Figure 6: here is the core finding for the authors' claim on the relationships between divergence time and selection pressure. It is clear that, using these species pairs, the earlier diverging (peak Ks, not sure why the more accurate divergence time estimate not used here) ones tend to have larger proportion of genes under stronger selection. The effect size of the correlation is rather remarkable. One unclear aspect of the analysis is whether the genes examined are the same across species pairs. It was noted that there are few orthologous genes that are 1-to-1 among most species pairs. Thus, it is likely the analysis is done using all available pairs. If that is the case, how should one compare one species pair to another?

The same genes may be under different selection pressures in different lineages. The group of genes under the same category of selection pressures may be different among taxon pairs. This does not prevent one to compare abundance of genes under different pressures and reported the observed pattern.

- Related to the point above, the patterns based on proportions are difficult to interpret. A direct way to demonstrate the authors' point is by examining the evolution rate of EACH orthologous gene pair and show that a gene experienced stronger purifying selection initially than later on the selection was relaxed. Since this is central to the authors' major point, this should be demonstrated. In addition to potentially providing support for the authors' claim, it will provide some resolution to the study in which genes contribute to the patterns, particularly if the authors follow up with gene set enrichment analysis on those exhibit the expected patterns.

The reviewer may have misunderstood the manuscript. We reported the pattern (the abundance of genes >0.1, <0.5 were positively related to the divergence time, and abundance of genes with *dN/dS*, 0.1 negatively related to divergence times) and attempted to speculate a plausible explanation for this pattern of early divergence of the gene. We thought it is indirect evidence suggesting a scenario about relaxing of selection pressure of genes as evolution proceed in time. However, this relaxing of selection pressure may occur in different genes independently in different taxon pairs. This scenario does not predict that each orthologous gene will relax the selection pressure. Nonetheless, it is interesting to examine some orthologous genes shared among taxon pairs since the *Ka/Ks* variation of a gene through time cannot be tracked within a species pair. We examined the *Ka/Ks* ratio for 79 orthologous genes shared in 90 or more taxon pairs. We did a scattered graph between the divergence times and the *dN/dS* values that are less than 0.5 for each of the 79 genes to see if there are genes showing a trend of positive relationship between the two variables. Among these genes, a majority of them do not show apparent trend of relationship (with R squares <0.05), while 21 of them show a trend of positive relationship and 16 showing a trend of negative relationship. The relationships are strong in a few genes in either pattern. The analyses were done by removing *dN/dS* values = 0 or 99, and those >0.5.

The observation of increase of the proportion of genes in the category of 0.1<*ks/Ks*<0.5 only suggests probably selection relaxation of some genes during this early diverging stage. Beyond this level of purifying divergence, we did not observe a strong relationship between time and gene enrichment.

We have revised the manuscript and added these analyses in. The observation of a negative relationship between *Ks* in peak abundance (reflecting time) and enrichment of genes with *Ka/Ks* >1 in our study was also reported in previous studies in mammals.

- Discussion section: The authors discussed a few of the caveats in the study which is helpful. Aside from the point on rate estimate raised earlier, I am not sure that I agree with the authors' assertion that "the likelihood of non-orthology of these genes is probably very low…" using transcriptome-based inference. The arguments are not supported by quantitative information and is hand-wavy. It will be useful to bring in comparative genomic studies as reference points. Otherwise, it remains a potential thorny issue in their methodology.

OrthoMCL performs well in identifying orthologous genes (Chen et al., 2007; Altenhoff and Dessimoz, 2009). We have reworded the text and cited references to support the argument. The transcript pairs were from the same tissue (leaf samples) collected in the same season. The assumption is that for recently diverged sister species, it is more likely that the same copy of genes are expressed in the same tissue during the same growing season as a result of gene functional conservation. We therefore cautiously call them putative orthologous single copy genes. Comparative genome sequences can confirm if the POGs are truly single copy. Only gene genealogy from phylogenetic analyses can confirm the orthology of genes from different taxa. Unfortunately, we do not have genome sequences for any pairs of taxon at the moment. Results of phylogenetic analyses of POGs shared among taxa showed all taxon pairs were grouped as sister taxa, suggesting that the sequence pairs are likely orthologous genes. Otherwise, we would have observed separation of sister taxa on the gene tree. We checked on the gene tree for each of the 7 genes shared by all 20 taxon pairs, and in all cases, the two sequences from the same taxon pair were grouped as sisters. These results provide evidence concordant with orthology of the sequence pairs. Most sequence pairs were highly similar (as evidenced by the low *Ks* values). We also did the same analyses for the 79 genes shared by 90% or more taxon pairs and found the same result. We believe the probability of non-homology of the sequence pairs is low. Comparative genome analyses of 20 angiosperms identified single copy orthologous genes.

Reviewer #3:The paper looks at the evolution of genes in pairs of species that have been isolated geographically. To do so, the authors selected 20 pairs that have a South East Asia versus North America disjunction and calculated the ratio of nonsynonymous versus synonymous substitutions across their transcriptomes. This is a clever approach. Strikingly, they found some sort of constancy in the results, with that >90% of the genes examined being under purifying selection and <10% being under positive selection. This is what we would expect, but it is a nice way to show it, using comparative analyses and large datasets. I would think people have looked at this in model organisms (mice, humans), but not throughout wild plant species. I think it would definitely make a great contribution to eLife, although I think the author could make more of a general point by adding other pairs of taxa, such as those from a North to South America disjunction, or South America versus Africa disjunction, or maybe even using species that have colonised island versus mainland sisters.

That would be a future study. Unfortunately, we do not have data for those pairs at the moment.

I think the text could be made clearer to the readers, e.g. explaining, what is the expectation of Ka/Ks ratio under positive or purifying selection (I would also use Kn rather than Ka for nonsynonymous substitutions).

We inserted new text in the Introduction to explain this. *Ka* is a standard symbol for rate of nonsynonymous substitutions in textbook.

I noted there are some custom scripts used here, have they been made available as supplementary information or in a public repository?

Yes, they are deposited on a website and the URL is provided in the method (https://github.com/ybdong919/GDMETS.git.). We made it clear in the text now and the script is available here.

For the dating exercise, did I understand correctly that the tree was built and dated with BEAST using 7 POGs shared by all taxa, and then the dates were plotted against Ks for the other genes to avoid circularity?

We also estimated divergence time with 79 genes in 90% or more taxon pairs.

Finally, I think the figures could be made much nicer (artwork); there are also many inconsistencies in the reference section; but I would recommend acceptance pending concerns above have been taken into account.

We have revised the figures and checked the references. Thanks.